# Continual Learning via Local Module Composition

**Oleksiy Ostapenko**[12]    **Pau Rodríguez**[3]    **Massimo Caccia**[123]    **Laurent Charlin**[145]
[1]Mila - Quebec AI Institute, [2]Université de Montréal, [3]ServiceNow, [4]HEC Montréal,
[5]Canada CIFAR AI Chair

## Abstract

Modularity is a compelling solution to continual learning (CL), the problem of modeling sequences of related tasks. Learning and then composing modules to solve different tasks provides an abstraction to address the principal challenges of CL including catastrophic forgetting, backward and forward transfer across tasks, and sub-linear model growth. We introduce local module composition (LMC), an approach to modular CL where each module is provided a local structural component that estimates a module's relevance to the input. Dynamic module composition is performed layer-wise based on local relevance scores. We demonstrate that agnosticity to task identities (IDs) arises from (local) structural learning that is module-specific as opposed to the task- and/or model-specific as in previous works, making LMC applicable to more CL settings compared to previous works. In addition, LMC also tracks statistics about the input distribution and adds new modules when outlier samples are detected. In the first set of experiments, LMC performs favorably compared to existing methods on the recent Continual Transfer-learning Benchmark without requiring task identities. In another study, we show that the locality of structural learning allows LMC to interpolate to related but unseen tasks (OOD), as well as to compose modular networks trained independently on different task sequences into a third modular network without any fine-tuning. Finally, in search for limitations of LMC we study it on more challenging sequences of 30 and 100 tasks, demonstrating that local module selection becomes much more challenging in presence of a large number of candidate modules. In this setting best performing LMC spawns much fewer modules compared to an oracle based baseline, however it reaches a lower overall accuracy. The codebase is available under `https://github.com/oleksost/LMC`.

## 1   Introduction

The goal of continual learning (CL) is to learn efficiently from a non-stationary stream of tasks without (catastrophically) forgetting previous tasks [62]. CL is often modeled as a trade-off between knowledge retention (stability) and knowledge expansion (plasticity) [26, 64]. Parameter sharing can provide control over this trade-off. For example, learning a single model shared across tasks results in better knowledge transfer and faster learning at the expense of forgetting [46, 57]. Conversely, learning a separate model per task eliminates forgetting but minimizes transfer and data efficiency [2, 41].

Modular learning aims at balancing transfer and forgetting by learning a set of specialized modules that can be recomposed to solve (new) tasks while only updating a subset of relevant modules or adding new modules [6, 47, 27]. In principle, a modular learner capable of composing modules in meaningful structures can provide additional benefits including **(i)** computational gains due to only executing modules that are relevant to a task [47, 4]; **(ii)** memory gains due to instantiating a sub-linear number of modules w.r.t. the number of tasks; **(iii)** systematic [8] and out-of-distribution (OOD) generalization [18] through knowledge recombination; and **(iv)** biological plausibility [91, 90, 96].

---

corresponding author: oleksiy.ostapenko@t-online.de

35th Conference on Neural Information Processing Systems (NeurIPS 2021).

Designing modular methods for CL comes with two main challenges. The first is *how and when to add new modules* to ensure sufficient plasticity to learn new tasks. Existing modular methods use greedy search variants, expanding the model when it improves validation performance [92, 63]. The second challenge is *how to compose* that is, retrieve task-specific structural knowledge given a new task (previously seen or not).

Existing methods rely on a task's identifier (ID) to retrieve task-specific structural knowledge, which comes either in the form of an optimal module layout [92] or as a model- and task-specific controller network that generates modular layouts [63]. Unfortunately, in many realistic CL scenarios task identities are unavailable at test time [23, 35, 14]. Lifting this limitation is challenging since standard mechanisms for task inference, for example, leveraging a task-inference model, could be subject to forgetting themselves.

To address both challenges, we equip each module with a *local structural component* that predicts a score indicating how relevant the module is for a given input. In-distribution inputs result in high scores, while out-of-distribution inputs result in low scores. In other words, modules self-determine their relevance given an input.

This local component is used for composing modules: for each datum, modules are combined at each layer according to their normalized scores without requiring a task's ID (§3). The local component is also used for module expansion: a new module is instantiated if all the current modules flag their input as being locally out-of-distribution (§3.1). Further, new shallow modules (i.e. closer to the input) are first trained in a *projection phase* to maximize the relatedness scores of subsequent, deeper, modules (§3.2). This process projects the output of new modules into the representation space expected by the subsequent modules and ensures the compatibility between low- and high-level modules.

In a set of studies, we explore the performance and versatility of our local structural approach, which we call Local Module Composer (LMC). First, we show that LMC reaches superior or comparable performance to existing modular and non-modular methods without requiring task IDs at test time using the Continual Transfer Learning (CTrL) benchmark, designed to evaluate transfer and forgetting in CL [92] (§4.1). Then, we demonstrate how LMC, relying on its projection phase, can solve out-of-distribution (OOD) tasks not seen during the continual training (§4.2). We also show it is possible to combine modules from independently trained models into a new model to solve tasks seen by each of the independent models without any finetuning (§4.3). Finally, an analysis of longer task sequences (30 and 100 tasks) reveals that LMC tends to spawn much fewer modules to reach good performance than the fully task-aware MNTDP [92] counterpart. However, LMC reaches slightly lower accuracy on longer sequences than MNTDP, which highlights the difficulty of automatic task-ID agnostic module selection in the presence of a large number of candidate modules. In Appendix F we demonstrate the applicability of LMC in the meta-continual learning (meta-CL) setting, a task-agnostic setting by nature.

We highlight that by relying on a local (per-module) structural component, LMC offers a modular CL approach that i) does not require task IDs during test in the standard task incremental settings; ii) balances parameter sharing to yield strong CL performances compared to baselines that require access to the task ID; iii) in our experiments instantiates a sub-linear number of modules; iv) permits recombination of modules at test time enabling OOD generalization as well as (v) the ability to combine independently trained models in a third model without fine-tuning. Notably, the OOD generalization is only possible if the agent is task-agnostic in the module selection process, since OOD tasks were not observed at training, the learner has to interpolate between the learned tasks, and a (categorical) task ID is of no use.

## 2   Background: Modular Continual Learning

Let $\mathcal{F}(x; \theta) : \mathcal{X} \rightarrow \mathcal{Y}$ be a learner parametrized with a set of parameters $\theta$. In task-incremental CL, the learner is exposed to a sequence of tasks. Each task is composed of a training set $D_t$ of $(x, y)$ pairs and a task identifier (ID) $t$ [92, 46]. The goal is to learn an optimal $\theta^*$ that minimizes the loss $\mathcal{L}$ for all observed tasks:

$$\theta^* = \arg\min_{\theta} \sum_{t=1}^{T} \mathbb{E}_{(x,y) \sim D_t} [\mathcal{L}(\mathcal{F}(x; \theta), y)]. \tag{1}$$

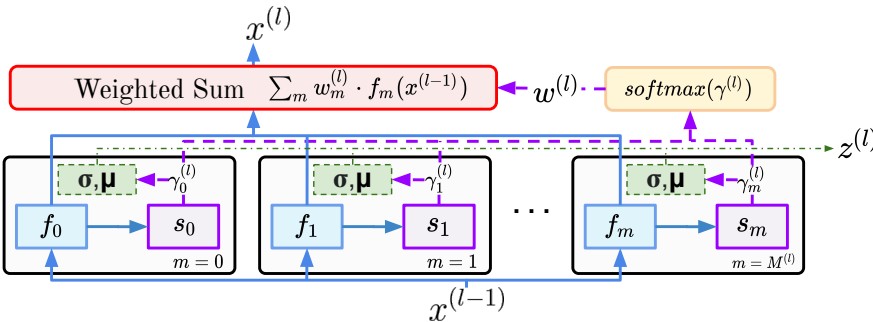

Figure 1: **Modular Layer Scheme**. Each black rectangle is a module. Inside each module, the functional component $f_m$ receives the input $x^{(l-1)}$ and feeds its output to the structural component $s$. The output of the structural component is used to calculate the importance score $\gamma_m^{(l)}$ using Eq. 5, which are normalized to obtain the attention vector $w^{(l)}$. The layer output is the weighted sum of the functional outputs of each module. $\mu$ and $\sigma$ are the running mean and variance of the scores $\gamma_m^{(l)}$ used to detect outlier inputs and to trigger module addition.

The parameter sharing trade-off between tasks can be addressed through different architectural design choices for $\mathcal{F}$. For example, $\mathcal{F}$ can be a monolithic network that shares parameters $\theta$ across all tasks. Most existing task incremental CL methods use a task-specific output head, requiring the task ID to select the output head corresponding to the task at hand [46, 87, 1].

At the other end of the spectrum are the expert based solutions that learn an independent model, a.k.a. expert, for each task [2, 83]. In this case, each expert trains task-specific parameters $\theta = \{\theta^{(t)}\}_{t=1}^T$.

To balance parameter sharing and transfer, modular methods organize their parameters in a series of modules $M = \{m_k^{(l)}\}$ with parameters $\theta = \{\theta_k^{(l)}\}$, where $\theta_k^{(l)}$ denotes the parameters of module $k$ at layer $l$ in $\mathcal{F}$. In general, a module can be any parametric function. In our experiments, unless otherwise stated, a module consists of a single convolutional layer followed by batch-norm, ReLU activation, and a max-pooling operation.

Modules can be composed conditioned on a sample, a batch of samples, or a task. Let $\psi$ denote a specific composition of modules that gives rise to a distinct prediction function; we make this dependence explicit: $\mathcal{F}(x; \theta, \psi)$. Importantly, sharing modules across tasks should lead to desirable transfer properties.

Veniat et al. [92] frames modular CL as finding an optimal layout $\psi^{(t)}$ for each task, where each layout selects a single module per layer per task (hard selection):

$$\theta^*, \Psi^* = \arg\min_{\theta, \Psi} \sum_{t=1}^T \mathbb{E}_{(x,y)\sim D_t}[\mathcal{L}(\mathcal{F}(x; \theta, \psi^{(t)}), y)]. \tag{2}$$

In this case the set of layouts $\Psi = \{\psi^{(t)}\}$ grows with the number of tasks, while modules can be reused across different task-specific layouts resulting in sub-linear growth pattern. They design a method called MNTDP to search the exponentially large space of modular layouts by only considering layouts resulting from adding a new module per layer to the best prior path (past task's path with the highest nearest neighbor accuracy on a new task) starting at the top layer. This solution relies on task IDs to retrieve $\psi^{(t)}$ at test time.

Another way of composing modules uses dynamic routing [63, 82, 47, 65]. The module layout is generated by a structural function $\psi = s(x)$, hence different inputs take different routes through $\mathcal{F}$. It is standard to approximate the structural function using a neural network $\psi = s(x; \phi)$ with structural parameters $\phi$. This framework has been applied to CL in [63] by learning a separate structural function per task $\psi^{(t)} = s(x; \phi^{(t)})$. The task IDs are used to retrieve the correct structural function:

$$\theta^*, \Phi^* = \arg\min_{\theta, \Phi} \sum_{t=1}^T \mathbb{E}_{(x,y)\sim D_t}[\mathcal{L}(\mathcal{F}(x; \theta, s(x; \phi^{(t)})), y)], \tag{3}$$

where $\Phi = \{\phi^{(t)}\}$ is the set of structural parameters for all tasks.

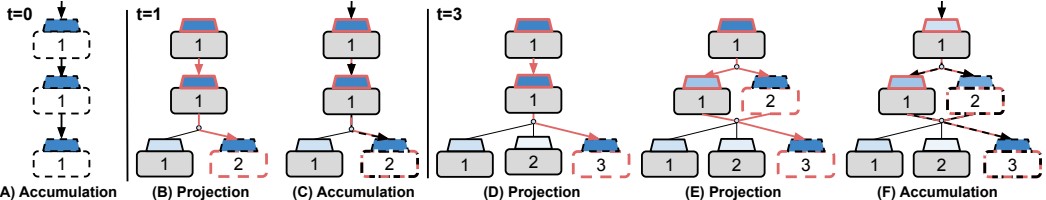

Figure 2: **Two-phase training**. Each module contains a functional (rectangle) and structural (trapezoid) component. Their color intensity denotes the strength of their activation. Components with dashed contours are trained, solid contours represent fixed components, arrows show the gradient flow: black arrow — *functional* signal, pink — *structural* signal. **(A)** All modules are trained on task 0. **(B)** Task 1 arrives, a new module is added at layer 1, which is first trained to project its output into the representation of the modules above via the *structural* signal (the *functional* signal is optional). No module addition is allowed during the projection phase. **(C)** Module addition is allowed again, both signals are used for training. **(D)** As task 3 arrives, a new module is added at layer 1 again, projection phase is triggered. **(E)** A new module is added at the layer 2, both new modules are now trained in the second projection phase. **(F)** Both new modules are trained using both signals.

The above methods require task IDs at both training and testing time. Next we introduce our modular CL approach that only relies on task IDs during training.

## 3 Local Module Composer (LMC)

We propose LMC, a modular approach where each module consists of a functional component $f(x; \theta_m^{(l)})$ and a structural component $s(x; \phi_m^{(l)})$, see Figure 1. The functional components are responsible for learning to solve the prediction task and are trained via the usual task loss $\mathcal{L}$ (e.g. cross-entropy loss for classification). The structural components receive the corresponding functional output as their input (see Figure 1) and are responsible for dynamic routing through $\mathcal{F}$. Structural parameters $\phi$ are trained using a structural loss $\mathcal{L}^{(st.)}$ computed locally at each module.

Intuitively, the structural component of a module should serve as a density estimator of the outputs of the functional component. The module's contribution to the layer's output is proportional to the likelihood of the input sample under the estimated density. In our instantiation, the structural component produces a relatedness score: a lower score for inputs that are more likely to belong to the distribution on which a given module was trained, and a higher score for inputs that are out-of-distribution for the given module. Hence, the likelihood of the input sample is approximated by the negative relatedness score.

Given an input data sample $x^{(0)} = x$, the output $x^{(l)}$ of a layer $l$ is defined as the weighted sum of the functional outputs of all $|M^{(l)}|$ local modules and used as input to the subsequent layer $l + 1$:

$$x^{(l)} = \sum_{m=1}^{|M^{(l)}|} w_m^{(l)} \cdot f(x^{(l-1)}; \theta_m^{(l)}). \tag{4}$$

The functional output of the network is equal to the output of the final layer: $\mathcal{F}(x; \theta, \phi) = x^{(L)}$. In the last layer $\mathcal{F}$ implements a single output-head per task. At training time, the task ID is available and we update only the output-head corresponding to the currently learned task. At test time, the task ID is not available and we select the output head with the highest activation weight $w_m^{(L)}$, i.e., the last layer performs hard module selection.

The module activation weights $w_m^{(l)}$ are computed by normalizing the vector of local relatedness scores $\gamma^{(l)} \in \mathbb{R}^{|M^{(l)}|}$. Each element of $\gamma^{(l)}$ is obtained from the negative structural loss which approximates the likelihood of each module:

$$\gamma_m^{(l)} = -\mathcal{L}^{(st.)}\Big(s\big[f(x^{(l-1)}; \theta_m^{(l)}); \phi_m^{(l)}\big]\Big), \tag{5}$$

$$w_m^{(l)} = \text{softmax}(\gamma^{(l)})_m. \tag{6}$$

Modules with lower structural loss get higher activation weights. Note that in practice, it can be useful to bias the module selection towards the expected module selection in a batch, assuming that samples within a batch are likely to belong to the same task. We discuss this point further in § A.1.

Instead of using the softmax function, it is possible perform hard selection taking the module with the highest score [82], or alternatively selecting top-k modules [88]. In both cases, LMC's structural parameters stay differentiable due to the local nature of structural learning. Note that in the case of global structural objective, hard module selection would require applying tools for non-differentiable learning such as Expectation Maximization [47] or reinforcement-learning based methods [82, 5].

The overall LMC objective consists of optimizing both functional and structural losses:

$$
\theta^*, \phi^* = \arg\min_{\theta,\phi} \sum_{t=1}^{T} \mathbb{E}_{(x,y)\sim D_t} \Big[ \mathcal{L}\big(\mathcal{F}(x;\theta,\phi),y\big) + \sum_{l=0}^{L} \sum_{m=0}^{|M^{(l)}|} \mathcal{L}_m^{(st.)}\big(s[f(x^{(l-1)};\theta_m^{(l)});\phi_m^{(l)}]\big) \Big].
$$
(7)

As in [92], learning is performed w.r.t. only newly introduced modules to prevent forgetting.

**Structural component.** We test two instantiations of the structural component $s$ and loss $\mathcal{L}_m^{(st.)}$. In the first one, $s$ is an invertible neural network [80]. Here we use the invertible architecture proposed by Dinh et al. [21]. As shown by Hocquet et al. [37], for this invertible architecture the structural objective can be defined as $\mathcal{L}_m^{(st.)}(x) = ||x||_2$. Intuitively, an invertible architecture prevents $\mathcal{L}_m^{(st.)}$ from collapsing to an all-zeros solution.

In the second instantiation, $s$ and $f$ form an autoencoder and $\mathcal{L}_m^{(st.)}(x) = ||x^{(l-1)} - x||^2$ is the reconstruction error with respect to the module's input $x^{(l-1)}$. Aljundi et al. [2] used a similar idea was for selecting the most relevant expert network conditioned on a task. Unless stated otherwise, modules in the feature extractor use the autoencoder as their structural component, while output heads use invertible $s$ — these combinations worked well in practice.

### 3.1 Expansion strategy

It is necessary to expand $\mathcal{F}$ as new tasks arrive to acquire new knowledge. A new module is added to a layer when all modules in this layer detect an outlier input. To this end, we track the running statistics of the relatedness score $\gamma$ for each module — mean $\mu$ and variance $\sigma$ (see Figure 1), and calculate a z-score for each sample in the batch and each module at a layer:

$$
z_m = \frac{w_m - \mu_m}{\sigma_m}.
$$
(8)

An input is considered an outlier if its z-score is larger than a predefined threshold $z'$ (see Appendix B.6 for an ablation study of $z'$ values). The expansion decision can be made on the per-sample (i.e., if an outlier sample is detected) or a per-batch basis (i.e., $z$ is averaged over the mini-batch). Unless stated otherwise, in our experiments, the decision was made on a per-batch basis. Additionally, at training the parameters of existing modules are fixed once the task changes. If during a forward pass through $\mathcal{F}$ module addition is triggered at multiple layers, we start adding modules at the layer closest to the input.

### 3.2 Training

Each module in LMC receives two types of learning signal: a structural signal resulting from minimizing $\mathcal{L}_m^{(st.)}$, and a functional signal resulting from minimizing the global functional loss $\mathcal{L}$. All structural components $s$ are trained *only* with the structural signal that is calculated locally to each module.

The training of functional components proceeds in two phases: projection and accumulation. Whenever the expansion strategy triggers the addition of a new module (i.e., $z_m > z' \ \forall m \in \{0, \ldots, |M^{(l)}|\}$), starting with layers closest to the input, LMC initiates the *projection phase*. During this phase, the new module is trained to minimize the structural loss from all the layers above and *no new-module addition is allowed*. This procedure makes the representation of new modules

| | $S^-$ | | | $S^+$ | | | $S^{in}$ | | | $S^{out}$ | | | $S^{pl}$ | | |
|---|---|---|---|---|---|---|---|---|---|---|---|---|---|---|---|
| MODEL | $\mathcal{A}$ | $\mathcal{F}$ | M | $\mathcal{A}$ | $\mathcal{F}$ | M | $\mathcal{A}$ | $\mathcal{F}$ | M | $\mathcal{A}$ | $\mathcal{F}$ | M | $\mathcal{A}$ | $\mathcal{F}$ | M |
| HAT[87] | 63.7±0.7 | -1.3±0.6 | 24* | 61.4±0.5 | -0.2±0.2 | 24* | 50.1±0.8 | 0.0±0.1 | 24* | 61.9±1.3 | -3.2±1.3 | 24* | 61.2±0.7 | -0.1±0.2 | 20* |
| EWC[46] | 62.7±0.7 | -3.6±0.9 | 24* | 53.4±1.8 | -2.3±0.4 | 24* | 56.3±2.5 | -9.1±3.3 | 24* | 62.5±0.9 | -3.6±0.9 | 24* | 54.2±3.1 | -4.2±2.7 | 20* |
| O-EWC[85] | 62.0±0.7 | -3.2±0.7 | 24* | 54.6±0.7 | -1.3±1.0 | 24* | 54.2±3.1 | -10.8±3.1 | 24* | 62.4±0.6 | -3.0±0.9 | 24* | 52.3±1.4 | -5.7±1.3 | 20* |
| ER[81, 16] | 60.6±0.7 | -2.1±0.9 | 4* | 63.0±0.6 | 3.8±0.8 | 4* | 63.8±1.4 | -1.9±0.6 | 4* | 60.7±1.0 | -1.5±0.5 | 4* | 60.5±1.0 | 0.5±0.9 | 4* |
| EXPERTS | 62.7±0.9 | 0.0 | 24 | **63.2±0.8** | 0.0 | 24 | 63.1±0.7 | 0.0 | 24 | 63.1±0.7 | 0.0 | 24 | **63.9±0.5** | 0.0 | 20 |
| MNTDP[92] | 66.3±0.8 | 0.0 | 13.7 | 62.6±0.7 | 0.0 | 21.0 | 67.9±0.9 | 0.0 | 16.0 | 65.8±0.9 | 0.0 | 15.0 | 64.0±0.2 | 0.0 | 17.2 |
| SG-F[63] | 63.6±1.5 | 0.0 | 14.7 | 61.5±0.6 | 0.0 | 20.8 | 65.5±1.8 | 0.0 | 17.5 | 64.1±1.3 | 0.0 | 16.2 | 62.0±1.3 | 0.0 | 16.0 |
| LMC(¬A) | **66.6±1.5** | -0.0±0.1 | 15.3 | 60.1±2.7 | -1.4±2.4 | 21.3 | **69.5±1.0** | 0.0±0.1 | 20.0 | **66.7±2.2** | -0.1±0.1 | 15.5 | 61.6±4.8 | -3.5±3.1 | 18.2 |
| MNTDP(A) | 41.9±2.5 | -2.8±0.6 | 14.8 | 43.2±1.3 | -10.8±2.0 | 20.7 | 32.7±13.6 | -15.2±13.2 | 17.2 | 37.9±2.7 | -5.8±3.5 | 13.3 | 35.1±3.6 | -16.4±4.6 | 15.8 |
| LMC(A) | **67.2±1.5** | -0.5±0.4 | 15.7 | 62.2±4.5 | 2.3±1.6 | 22.3 | **68.5±1.7** | -0.1±0.1 | 19.7 | **55.1±3.4** | -7.1±4.0 | 15.5 | **63.5±1.9** | -1.0±1.5 | 19.0 |
| LMC(A,H) | 64.9±1.5 | -0.2±0.2 | 16.2 | 55.8±2.5 | -0.3±1.2 | 15.3 | 67.6±2.7 | -0.8±1.0 | 21.5 | 54.2±3.6 | -2.9±2.0 | 15.9 | 53.8±5.7 | 3.1±5.5 | 10.8 |
| SG-F(A) | 29.5±3.5 | -35.3±4.0 | 14.3 | 20.4±4.4 | -39.3±6.7 | 16.0 | 24.4±5.6 | -38.7±4.0 | 18.7 | 30.5±4.5 | -34.0±5.5 | 12.2 | 19.4±1.0 | -41.8±1.6 | 15.5 |
| ER(A,S)[81, 16] | 60.4±1.0 | -0.5±0.7 | 4* | **65.3±0.9** | 6.0±1.0 | 4* | 58.8±3.2 | -4.2±3.7 | 4* | 47.6±1.5 | -7.6±1.6 | 4* | 58.6±1.3 | -1.2±1.5 | 4* |
| FINETUNE | 47.5±1.5 | -14.9±1.4 | 4* | 31.4±3.7 | -29.3±3.8 | 4* | 39.7±5.0 | -23.9±5.7 | 4* | 45.4±4.0 | -15.5±3.7 | 4* | 29.1±3.1 | -29.2±3.2 | 4* |
| FINETUNE L | 52.1±1.4 | -15.7±1.7 | 24* | 38.2±3.2 | -25.8±3.3 | 24* | 49.3±2.0 | -18.4±2.0 | 24* | 49.3±2.1 | -18.4±2.0 | 24* | 37.1±2.1 | -26.0±2.2 | 20* |

Table 1: **CTrL results:** we report accuracy ($\uparrow \mathcal{A}$), forgetting ($\uparrow \mathcal{F}$) with standard deviations calculated over 6 different runs. We report the mean number of modules (M) over these runs, where * marks methods with fixed capacity. The first block comprises a set of standard CL baselines including regularization and replay based methods. The second block are the modular methods, third – modular and replay based methods that are task ID agnostic (A), and the last block are the two finetuning baselines. (H) indicates hard module selection. (S) indicates single-head as detailed in the main text.

compatible with subsequent modules and enables their composition. This procedure "encourages" already-learned modules to be reused, preventing over-spawning new modules. The functional signal is optional during projection (we kept it in all experiments unless otherwise stated).

In the *accumulation phase*, new module addition is allowed again and all non-frozen modules are trained with both signals. The functional components of new modules still receive a signal from the structural components of modules above. The two-phase training is explained schematically in Figure 2 and implementation details are provided in Appendix A.

## 4 Experiments

We now evaluate the performance, empirical capabilities, and properties of LMC in four different CL settings. First, in § 4.1 we study a standard task-incremental CL setting (task-ID agnostic and aware) using the Continual Transfer Learning Benchmark (CTrL) [92]. Next, we explore the properties of LMC through other CL settings. In § 4.2 we evaluate the continual OOD generalization ability of the proposed LMC. In § 4.3 we show the ability of LMC to combine modules form independently trained models. In Appendix F, we evaluate LMC in the Continual Meta-Learning setting.

### 4.1 Continual transfer learning using the CTrL benchmark

The CTrL benchmark was proposed to systematically evaluate properties of CL methods with a focus on modular architectures [92]. It consists of 5 streams of visual image classification tasks. The first stream $S^- = (t_1^+, t_2, t_3, t_4, t_5, t_1)$ consists of a sequence of 6 tasks, where the first and last task are the same except the first has an order of magnitude more training samples ("$+$") than other tasks. This stream is designed to evaluate the *direct transfer* ability of models, i.e. a modular learner should be able to *reuse* the first task's modules for the last task. The $S^+ = (t_1, t_2, t_3, t_4, t_5, t_1^+)$ stream is similar to $S^-$, but now the last task comes with more data than the other tasks (including the first one). Here, the modular learner should be able to *update its knowledge*, i.e. performance on the first task should improve after learning the last task. In the $S^{in} = (t_1, t_2, t_3, t_4, t_5, t_1')$ stream the first $t_1$ and the last $t_1'$ tasks are similar, with a slight input distribution change (e.g. different background color). In the $S^{out} = (t_1^+, t_2, t_3, t_4, t_5, t_1'')$ stream the first task $t_1$ and the last task $t_1''$ differ in the amount of training data and the output distribution, i.e. the labels of the last task are randomly permuted. The *plasticity* stream $S^{pl} = (t_1, t_2, t_3, t_4, t_5)$ evaluates the ability to learn a stream of unrelated and

potentially interfering tasks, i.e., transfer from unrelated tasks can harm performance. Descriptive statistics for all datasets are in Appendix B.1.

We compare to several baselines. **Finetune**: trains a single model (wider model marked with **L**) for all tasks. **Experts**: trains a model per task. We also compare with the several recently proposed modular CL baselines, which achieve competitive results in CTrL and require task IDs at test time. **MNTDP** [92]: a recent search-based module selection approach described in more detail in § 2. MNTDP requires the task ID to retrieve the previously found best structure for each test task. **MNTDP(A)**: a task ID agnostic version of MNTDP we created, which selects the path with the lowest entropy in the output distribution. **SG-F** [63]: Soft-gating with fixed modules, a modular method mentioned in § 2. It relies on a task-specific structural network that generates soft-gating vectors for each layer of the modular learner and fixes learned modules when new tasks arrive. We slightly adapted the original expansion strategy of SG-F in order to conform to our experimental setup; details are in Appendix B.2. **SG-F(A)**: a version of SG-F with a single structural network shared across tasks. **HAT**[87]: learns attention masks for activations that gate the gradients to prevent forgetting. The task ID is used to select a task-specific attention mask for inference.

We also compare to several standard CL methods. **EWC** [46]: trains a single model for all tasks while applying parameter-regularization to minimize forgetting. **O-EWC**:[85] online version of EWC that does not require storing a separate approximation of the Fisher information matrix per task. **ER** [16]: trains a single model while replaying samples from previously seen tasks. The size of the replay buffer corresponds to the memory size of the LMC assuming the worse case linear growth pattern (i.e., LMC with 24 modules on a 6-task sequence). **ER(A,S)**: task ID agnostic version of ER that uses a single output head to classify all classes from all tasks: i.e. after learning stream $S^-$ the output head has 50 output neurons and the output classes of the last task are considered the same as the ones of the first task.

We use several versions of LMC. **LMC(¬A)** a version of LMC that uses the task ID for output head selection (not module selection as MNTDP). **LMC(A)**: the default version of LMC. It equips output heads with structural components and is therefore task ID agnostic at test time. **LMC(A,H)**: a version of task ID agnostic LMC that performs hard module selection, i.e., taking the module with the highest relevance score per layer. All methods use the same architecture (described in Appendix A.4) together with the Adam [44] optimizer. HAT is the only method that uses SGD.

Similar to Veniat et al. [92], we use the following evaluation metrics: ($\mathcal{A}$) **average accuracy** on all seen tasks at the end of CL training; **Forgetting** ($\mathcal{F}$) — difference between accuracy at the end of the training and accuracy after learning the task averaged across tasks [60]; **Number of modules** (M) at the end of the continual training procedure. Formal definitions of all metrics are in Appendix B.3.

Table 1 reports performance using the CTrL benchmark. Overall, modular methods tend to outperform ER and the regularization based methods (HAT and EWC). Among the modular methods, soft-gating SG-F(A,F) with a single controller shared among all the tasks performed the worst. This baseline showcases the problem of forgetting in the global structural component (a.k.a. controller) of dynamic routing methods such as the one proposed by Mendez and Eaton [63]. A version of LMC performs the best on the $S^-$, $S^{in}$ and $S^{out}$ streams.

Notably, LMC(A), which does not rely on task IDs at test time, outperformed all other task ID agnostic methods such as MNTDP(A) and ER(A,S) on all streams but $S^+$, and always performed on par with task ID aware methods. On the $S^{out}$ stream low performance is expected for task-ID agnostic methods due to output distribution shift: i.e., at test time we notice that LMC correctly assigns samples from the last task $t_1''$ to the first task's $t_1^+$ output head. However, the resulting classification accuracy is low because the labels of the last task are randomly permuted in this stream.

The task ID agnostic LMC(A) outperforms task ID aware LMC on the $S^-$ and $S^+$ streams. Here, LMC(A) selects modules (and the output head) which were predominantly trained on the task that provided more training data (e.g. $t_1^+$ in $S^-$ stream), hence transferring knowledge between the first and the last tasks. In contrast, LMC(¬A) when tested on the last task $t_1$ is forced to select the output head belonging to this task, which was trained on less data than the output head of $t_1^+$ task, leading to lower accuracy. In addition, we observed that versions of LMC often exhibit high variance (e.g. see $S^+$, $S^{out}$ and $S^{pl}$ streams). This may be caused by the larger amount of trainable parameters compared to other models and relatively small amount of training data. Finally, low performance of

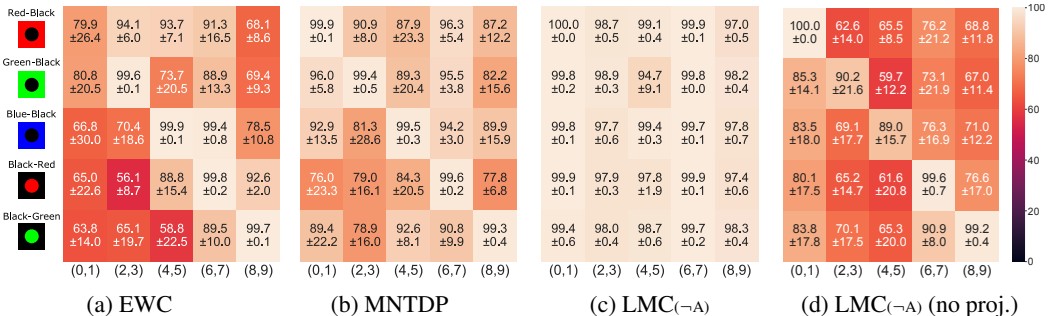

Figure 3: Continual OOD-generalization: matrices show test accuracy on seen and unseen tasks (on- and off-diagonal tasks respectively). In each sub-figure x-axis shows the MNIST class-combination used to build the task, y-axis gives the foreground-background colors. Only diagonal tasks are learned continually. While non-modular EWC (a) and modular MNTDP (b) can prevent forgetting, LMC(¬A) (c) is able to generalize to OOD tasks as well. LMC(¬A) without projection phase performs poorly as shown in (d).

LMC(A,H) emphasizes the importance of soft modular attention for LMC. Additional results, including a transfer metric [92], are in Appendix B.4.

## 4.2   Compositional OOD generalization

This second study tests the ability of LMC to recombine modules for OOD generalization. We use a colored-MNIST dataset — a variation of the standard MNIST dataset of hand-written digits from 0 to 9 [43] in which digits are colorized. We design a simple sequence of tasks as follows. First, we define two high-level features: the foreground-background color combination (using the colors red, black, green, blue) and the class (0–9). Then, we create five non-overlapping tasks of two (digit) classes each: {0-1, ..., 8-9}. At training time the model is continually trained using a sequence of these tasks, however, each task is only seen in one of five different foreground-background combinations {red-black, green-black, blue-black, black-red, black-green}. At test time we measure the generalization ability to seen and unseen combinations of classes and colors.

In Figure 3 we present the accuracy matrices for different learners when tested on all 25 combinations of colors and classes after it has been trained only on the 5 tasks on the diagonal.

We compare the performance of LMC(¬A) with EWC [46], MNTDP [92], and an ablated version of LMC without the projection phase. We observe that the OOD accuracy attained by LMC is significantly higher than EWC and MNTDP. Since the model trained with EWC is monolithic, the digit-background color combinations are entangled with the digits' shape for each task, hindering OOD generalization. In contrast, modular approaches such as MNTDP and LMC learn a different module combination for each task. In contrast to MNTDP, LMC's module selection does not rely on task identifiers and each module is selected in a local manner based on its compatibility with the current input. This allows LMC to interpolate between previously seen tasks being able to dynamically compose existing modules to adapt to tasks that have not been seen at training. Because MNTDP's module selection relies on a database of task-specific structures found to be optimal for the corresponding task at training, this method must reuse the predefined module compositions based on task IDs. This forces MNTDP to use modules that were trained using a different color combination, and results in e.g. a 24% accuracy drop with respect to LMC on [0,1] when the foreground and background colors are inverted w.r.t. the seen combination.

In Figure 3d we report results for LMC without applying the projection phase. The projection phase adapts the representation of newly introduced modules to match the distribution expected by the subsequent modules. As expected, we found that skipping it severely degrades performance. This result validates the usefulness of the projection phase to achieve an efficient local module selection. In Appendix E we plot the average module selection for all 25 test tasks, showing how modules are reused for the OOD tasks.

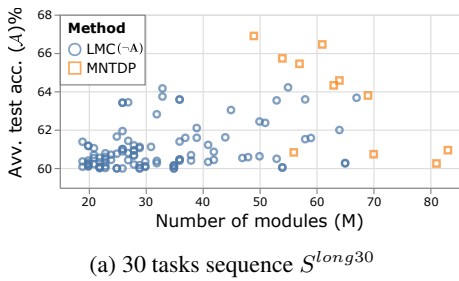
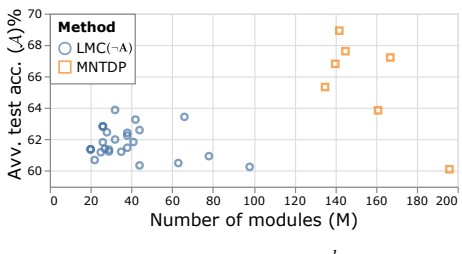

(a) 30 tasks sequence $S^{long30}$    (b) 100 tasks sequence $S^{long}$

Figure 5: Results on $S^{long}$ and $S^{long30}$ sequences for different hyperparameter values (we select only runs with reasonably good performance, i.e. $\mathcal{A}{>}60\%$), same plots plots for all conducted runs can be found in Appendix C).

### 4.3 Combining modular learners

In earlier sections we show cross-task reusability of modules, here we test the cross-model reusability. We motivate the practical importance of this kind of reusability with a federated learning example: a privacy preserving training might be required for LMC1 and LMC2, trained on the premises of customers 1 and 2, after which their modules can be combined in a single central entity — LMC3, located on premises of the cloud service provider. LMC3 is required to perform tasks seen by both independent LMCs but can not be finetuned as it has no access to the original training data distributions.

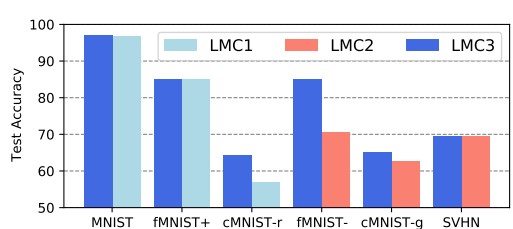

Figure 4: Performance of combining independently trained LMC1 and LMC2 into LMC3.

In Figure 4 we test the ability of LMC to preserve and transfer knowledge in such setting. To this end, we design the following tasks: fMNIST+ and fMNIST-. Both are sampled from the fashion-MNIST dataset [98] but the latter comes with an order of magnitude less training data. cMNIST-**r** is a variant of the colored-MNIST dataset where the background of 95% of the training samples is colored in **r**ed and 5% in **g**reen. For the cMNIST-**g** dataset these proportions are inverted and 95% of the training samples is colored in **g**reen. The test set contains 50% of samples with green and 50% with red background. We trained LMC1 continually on MNIST, fMNIST+, and cMNIST-r tasks. We trained LMC2 on fMNIST-, cMNIST-g, and SVHN. We then combined the modules of both LMCs layer-wise to obtain LMC3.

We observed positive transfer for both cMNIST and fMNIST- tasks with LMC3. We found that LMC3 selects different modules originating from different LMCs conditioned on test samples with different background colors — LMC1's modules were specialized on red background while LMC2's on green (selected paths presented in Appendix D). Notably, cross-model reusability without fine-tuning is novel to LMC and can be attributed to the local nature of the structural component. Using a global structural component as in [63] would require tuning a separate structural component specifically for LMC3. In case of task-specific routing of proposed for MNTDP [92], additional search would be needed to discover task-specific paths through the consolidated LMC3 model. In both cases the access to the orinal training data distributions would be required and privacy would not be preserved.

### 4.4 Longer task sequences

Here we study the performance LMC on longer task sequences consisting of 30 – $S^{long30}$, and 100 – $S^{long}$ tasks. The $S^{long}$ sequence corresponds to the one proposed by Veniat et al. [92] as part of the CTrL benchmark. $S^{long30}$ is a 30-tasks subset of $S^{long}$ (see Appendix B.1 for details).

We first report the average test accuracy ($\mathcal{A}$) and the total number of modules (M) of the models selected through cross-validation. $S^{long30}$: MNTDP $\mathcal{A}$=64.58, M=64; LMC$_{(\neg A)}$: $\mathcal{A}$=62.44, M=50. $S^{long}$: MNTDP $\mathcal{A}$=68.92, M=142; LMC$_{(\neg A)}$: $\mathcal{A}$=63.88, M=32. While the gap between the accuracy

of LMC and MNTDP on $S^{long}30$ is only 1.86%-points, in the case of $S^{long}$ this gap grows to 6.58%-points. It is important to highlight that in contrast to LMC, MNTDP's module selection is performed by a task ID aware oracle. We further analyze the trade-off between the number of modules and accuracy in Figure 5, where we plot the number of modules (M) against average test accuracy ($\mathcal{A}$) for models that resulted from training with different hyperparameters. For both streams, we observe that LMC tends to spawn much fewer modules than MNTDP. However, MNTDP shines in the presence of large number of modules and achieves higher overall test accuracy on these streams. Interestingly, as can be clearly observed on the $S^{long}$ stream, LMC reaches higher accuracy with smaller number of modules: e.g. $\sim$64% with 32 modules, while adding modules leads to lower accuracy: e.g. $\sim$61% with 98 modules. This result suggests that local task ID agnostic module selection becomes more challenging for LMC in presence of a large number of modules.

## 5 Related work

**Modularity in neural networks** is studied in the context of scalability [9], and more recently as a way to achieve compositionality and systematic generalization [6, 47, 15, 8, 27, 19] as well as for multi-task learning [65, 82]. From the causal point of view, a data generation process could be thought as a composition of independent causal modules [75]. Researchers model these kinds of systems using a set of independent modules, where each module is invariant to changes in the other modules induced by e.g. distribution shifts [84, 76]. This idea is crystallized by Parascandolo et al. [74], who propose a way to learn a set of causal independent mechanisms as mixture-of-experts. Building up on this ideas, others show evidence of compositional OOD generalization [61]. Recently, [66] argue for a more wholistic view on CL including OOD generalization as an important desiderata.

**Continual learning** methods typically address the problem of forgetting through parameter regularization [46, 70, 99], replay [89, 78, 3, 73, 54, 12, 97, 39, 81, 13] or dynamic architectures (and MoEs) [83, 87, 52, 85, 53, 41]. Our work falls under the umbrella of the latter and shares its advantage of having the capacity to adapt to a large number of related tasks. Our focus is on improving **modular CL** approaches, which despite their advantages, have only recently been studied in the CL literature [63, 92]. The main difference with our work is that we use a local composition mechanism instead of a global one. We detail this difference in §2 and also compare to these methods in §4.1.

**Continual-meta learning** focuses on fast learning and remembering [25, 35, 33, 41], often emphasising the online performance on OOD tasks [14]. As argued by Jerfel et al. [41] modularity can be useful in this setting to minimize interference between tasks. They proposed a way to train a MoE model, with each expert focusing on a cluster of tasks leveraging Bayesian nonparametrics. LMC aims at decomposing knowledge into layer-wise composable modules further reducing modular granularity. Continual-meta learning is often confused with its counterpart *meta-continual learning* [40, 11, 93], in which algorithm are learning to continually learn.

The rapid growth of continual learning has lead researchers to work on empirical studies [20, 58, 56], surveys [32, 42, 55, 66, 67] as well as CL-specific software [72, 22, 59].

## 6 Conclusion

We develop LMC, a method to learn and compose a series of modules on a continual stream of tasks fulfilling some of the basic desiderata of modular CL such as module specialization, avoidance of collapse, and sublinear growth. In LMC, structural information is learned and stored locally for each module. It is the locality of the structural component that enables generalization to related but unseen tasks, and that permits combining different LMCs without fine-tuning.

Future work could focus on achieving more efficient sub-linear model growth through OOD generalization and reusability of modules. Additionally, while the benefits of modularity for CL are well understood, the implications of the CL regime on modularity and compositionality have not been studied extensively. It is possible that providing knowledge to the learner in incremental chunks results in the implicit supervision needed to better disentangle it into specialized and composable modules. Another promising direction is removing the need for task boundaries during training and developing more robust architectures for the local structural component (related discussions are in Appendix A.3, and limitations in Appendix G).

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

## Acknowledgments and Disclosure of Funding

Laurent Charlin holds a CIFAR AI Chair Program and acknowledges support from Samsung Electronics Co., Ldt., Google, and NSERC. Massimo Caccia was supported through MITACS during his part time employment with Element AI the ServiceNow company. Massimo Caccia was also supported by Amazon, during his part time employment there. We would like to thank Mila and Compute Canada for providing computational resources. We also would like to thank Irina Rish for useful discussions.

