# A  Implementation and Algorithm details

## A.1  Batched modularity

Activation weights $w_m^{(l)}$ can be calculated separately for each data point. However, in batched regimes sometimes it can be assumed that module selection for samples in the same batch is likely to be similar.[1] This can be incorporated into LMC by redefining $w_m^{(l)}$ as:

$$w_m^{(l)} = \frac{1}{Z(w)}\text{softmax}(\gamma^{(l)}, \tau)_m \cdot \mu_{w,m}^{(l)}, \tag{9}$$

$$\mu_{w,m}^{(l)} = \mathbb{E}_{x_i \in b}\big(\text{softmax}(\gamma^{(l)}, \tau')_m\big), \tag{10}$$

where $Z(w)$ is the normalization term, $b$ denotes a batch of samples, $\tau$ and $\tau'$ are the temperature hyperparameters. Lower $\tau'$ would results in a stronger bias towards the expected selection $\mu_w^{(l)}$.

## A.2  Training with projection phase

The intuition behind the modular training with projection phase is the following: every time a new module addition is triggered (using the mechanism proposed in § 3.1), we start by only adding modules on the deepest layer — i.e. the one closest to the input. Then, in the *projection phase*, we train this module for some time using the signal coming from the structural components of the modules above (possibly combined with the training signal of the downstream task). Projection phase makes sure that the learner first tries to efficiently *reuse* the existing modules (the once above the newly added one) by trying to project it's output into the representation space expected by those modules. After some time the learner is allowed to add new modules again. If the previously added module was not enough to incorporate the distribution shift that caused the previous module addition, new module addition will be triggered in the layers above. We detail this procedure in the Algorithm 1 and 2. Additionally, for modules that recognize current input as outlier in the forward pass, we set their contribution for the current batch to zero (ll.14 in Algorithm 2). This ensures that the newly added modules get enough training signal to learn.

---

[1] E.g in case of locally stationary data distribution, samples seen together are likely to belong to the same task

**Algorithm 1:** Modular training with projection

1 **Require:** $k$ projection phase length, stream $S$, $z$-score threshold $z'$
2 **Initialize** Learner $\mathcal{F}_{\theta,\phi}$
3 **for** $t = 0...|S|$ **do**
4 $\quad D_t \leftarrow$ get dataset for task $t$
5 $\quad$ **for** $e = 0 ...$ *total epochs* **do**
6 $\quad\quad$ **foreach** *mini-batch* $b \in D_t$ **do**
7 $\quad\quad\quad X, y \leftarrow$ mini-batch of samples
8 $\quad\quad\quad \hat{y}, \ell^{str.} \leftarrow$ Forward Model$(\mathcal{F}_{\theta,\phi}, X, z', t)$
9 $\quad\quad\quad$ **if** *new module added in last k epochs* **then**
$\quad\quad\quad\quad$ // projection phase
10 $\quad\quad\quad\quad \ell = \ell^{(str.)}$
11 $\quad\quad\quad\quad$ **if** *use functional loss in projection* **then**
12 $\quad\quad\quad\quad\quad \ell += \mathcal{L}^{(fnc.)}(\hat{y}, y)$
13 $\quad\quad\quad$ **else**
14 $\quad\quad\quad\quad \ell = \mathcal{L}^{(fnc.)}(\hat{y}, y) + \ell^{(str.)}$
15 $\quad\quad\quad$ Update parameters $\theta, \phi$ using $\nabla_{\theta,\phi}\ell$
16 $\quad\quad$ **end**
17 $\quad$ **end**
18 $\quad$ **foreach** *module* $m \in \mathcal{F}$ **do**
19 $\quad\quad$ Fix structural parameter $\phi$ of module $m$
20 $\quad$ **end**
21 **end**

---

**Algorithm 2:** Forward Model

1 **Require:** Learner $\mathcal{F}_{\theta,\phi}$ with $L$ layers, batch $X$, $z$-score threshold $z'$, task index $t$
2 **Output:** logits $\hat{y}$, structural loss $\ell^{(st.)}$
3 $x^{(0)} = X$
4 $\ell^{(str.)} = 0$
5 **for** $l \in L$ **do**
6 $\quad$ Let $M^{(l)}$ denote a set of modules at layer $l$
7 $\quad$ Calculate:
8 $\quad \gamma^{(l)} \in \mathbb{R}^{|M^{(l)}| \times |X|}$ using Eq. 5,
9 $\quad w^{(l)} \in \mathbb{R}^{|M^{(l)}| \times |X|}$ using Eq. 9 or 6,
10 $\quad z^{(l)} \in \mathbb{R}^{|M^{(l)}| \times |X|}$ using Eq. 8, average over $|X|$
11 $\quad x^{(l)}$ using Eq. 4
12 $\quad$ **if** $t > 0$ **then**
13 $\quad\quad$ **foreach** $m \in M^{(l)}$ **do**
14 $\quad\quad\quad$ **if** $z_m^{(l)} > z'$ **then**
15 $\quad\quad\quad\quad$ **if**
$\quad\quad\quad\quad$ *module was added at layer l during task t*
$\quad\quad\quad\quad$ **then**
16 $\quad\quad\quad\quad\quad w_m^{(l)} = \vec{0}$ // does not use outlier modules
17 $\quad\quad\quad\quad$ **else if**
$\quad\quad\quad\quad$ *no module added in last k epochs*
$\quad\quad\quad\quad$ **then**
$\quad\quad\quad\quad$ // not in the projection phase
18 $\quad\quad\quad\quad\quad$ Fix all modules at layer $l$
19 $\quad\quad\quad\quad\quad$ Add a new free module to layer $l$
20 $\quad\quad\quad$ **end**
21 $\quad\quad \ell^{(str.)} = \ell^{(str.)} - \sum_m sum\{\vec{\gamma}_m^{(l)} \odot \vec{w}_m^{(l)}\}$
22 $\quad$ **end**
23 $\hat{y} = x^{(L)}$
24 **return** $\hat{y}, \ell^{(str.)}$

## A.3 Structural component

In practice, we applied the $log$-operation to the structural loss for both choices of the structural component, which resulted in a more stable training procedure.

### A.3.1 Invertible architectures

Invertible architectures, such as the one proposed by Dinh et al. [21], can be used to model high-dimensional density after mapping the data in a space with some desirable factorization properties. We use this idea here to directly approximation of the activation likelihood of a module $m$. More specifically, [37] show that maximizing the likelihood of a module under such invertible transformation is equivalent to minimizing the $L_2$ norm of the output of structural component $s(o; \phi_m)$, yielding the local structural objective:

$$\mathcal{L}_m^{(str.)}(x) = ||x||^2. \tag{11}$$

To satisfy the invertibility constraint [21] propose to split the input $o$ into blocks of equal size $o_1$ and $o_2$ and apply two, not necessarily invertible, transformations $s_1$ and $s_2$ as:

$$\begin{aligned} a_1 &= s_1(o_2; \phi_{2,m}) + o_1, \\ a_2 &= s_2(a_1; \phi_{1,m}) + o_2. \end{aligned} \tag{12}$$

The output of structural component $a$ is obtain through the concatenation of $a_1$ and $a_2$. Importantly, the input and output of the structural component have the same dimensionality $a_m, o_m \in \mathbb{R}^k$. The inverse can be obtained as:

$$\begin{aligned} o_2 &= a_2 - s_2(a_1; \phi_{2,m}), \\ o_1 &= a_1 - s_1(o_2; \phi_{1,m}). \end{aligned} \tag{13}$$

Intuitively, the invertibility constraint prevents $a_1$ and $a_2$ from collapsing to the solution of outputting 0-vectors, which would be useless.

### A.3.2 Other possible choices of structural component

The role of structural component in LMC is to detect in-distribution and out-of-distribution samples for each module. It is natural to consider density estimates produced by deep generative models (DGM) for this task. In this work we only considered a simple encoder-decoder based architecture and a simple flow-model. Applying other more complex DGMs such as VAEs [45] or flow-based [10, 48] models might further improve the efficacy of local structural component. Nevertheless, such models also come with their challenges, which include calibration difficulties as well as low data efficiency [95].

### A.4 Architecture details

Unless otherwise stated, we initialize the learner with a single module per layer, each learned consist of 4 layers. The used architecture of each module is detailed in Table 2. For the CTrL experiments we used invertible structural component for the task-specific output heads (classifiers) in task ID agnostic LMC(A), while for feature extracting trunk we used an encoder-decoder architecture (i.e. structural component tasked to reconstruct the module's input).

Unless stated otherwise, architectures used for all baselines closely resemble the architecture used by LMC. Thus, by default all baselines contain 4 layers with each layer's architecture and parameter count being equivalent to the architecture and the parameter count of the functional component of an LMC module. In modular baselines (MNTDP, SG-F) each module corresponds to the functional component of the LMC's module. Some of the fixed capacity baselines in Table 1 (e.g. HAT, EWC, Finetune L) where initialized with the layer width scaled up to match the parameter count of the largest possible modular network (e.g. in case of linear growth with one new module per layer per task the largest possible modular network in our framework would have 24 modules in a 6 tasks sequence).

| Functional component | | | | | | | | Structural component (decoder) | | | | | | |
|---|---|---|---|---|---|---|---|---|---|---|---|---|---|---|
| Type | |layer $\mathcal{F}$| | |layer m| | #params. | #out ch. | stride | padding | kernel | Type | |layer $\mathcal{F}$| | |layer m| | #params. | #out ch. | stride | padding | kernel |
| Conv. | 0 | 0 | 1792 | 64 | 1 | 2 | 3 | ConvTranspose2d | all | 0 | 16448 | 64 | 2 | 2 | 2 |
| Conv. | 1-3 | 0 | 36928 | 64 | 1 | 2 | 3 | Batch norm | all | 1 | 128 | - | - | - | - |
| Batch norm | all | 1 | 128 | - | - | - | - | ReLu | all | 2 | - | - | - | - | - |
| ReLu | all | 2 | - | - | - | - | - | Conv. | all | 3 | 65600 | 3 | 1 | 1 | 3 |
| Max. Pool | all | 3 | - | - | - | 2 | 2 | Sigmoid | all | 4 | - | - | - | - | - |

| Structural component (invertible) | | | | | |
|---|---|---|---|---|---|
| Type | layer $\mathcal{F}$ | |layer m| | #params. | #input | #output |
| Input $L_2$ Norm. | output head | 0 | - | - | |
| Linear | output head | 1 | 83232 | 288 | 288 |
| Linear | output head | 1 | 83232 | 288 | 288 |

Table 2: Used architectures per module: column 'layer $\mathcal{F}$' refers to the index of the layer in the modular learner $\mathcal{F}$ (i.e. 0 is the closes layer to the input), while 'layer m' gives the layer index within the module. Note, both linear layers in (c) are used in parallel as proposed in [21] (i.e. $s_1$ and $s_2$ from Eq. 13).

### A.5 Dealing with batch-norms and data normalisation

While batch normalization [38] is a useful device for accelerating the training of neural networks, it comes with challenges when it comes to settings with shifting data distribution such as meta- [7] and continual learning. Specifically, in continual learning when testing on the previously seen tasks after new tasks have been learned, the batch norm will change its statistics resulting in forgetting even if the parameters of the network have not been changed [31]. We highlight several ways to deal with it. One way is to warm-up batch norm before testing on previous tasks by performing several forward passes through the model with unlabeled test data to let batch norms "relearn" the task statistics. This assumes that the test data is available in high quantity at test time (i.e. we cannot warm-up batch-norm with a single test sample). Another way is to fix batch norms completely after a task has been learned. In monolithic architectures, such fixing might limit the plasticity of the network and prevent the learning of new tasks. In modular architectures, however, batch norms of frozen modules

can be kept frozen while new module's batch norms can keep learning resulting in a balance between stability and plasticity. In monolithic architecture which are task-ID aware at test time, a separate batch-norm layer can be used per task.

In modular methods we fixed the batch norms whenever the modules are fixed (e.g. after learning a task in LMC and MNTDP). In HAT we used a separate batch-norm per task, as using a single batch-norm resulted in high forgetting rates. For other non-modular methods (e.g. EWC, ER) we used a single batch-norm layer for all tasks. Additionally, we observed that monolithic methods that share batch-norms across tasks result in high forgetting rate if no data normalisation is performed (and no batch-norm warm-up before testing). In the CTrL experiments we normalised tasks' data for all methods but LMC using statistics calculated on each task separately. In these experiments LMC performed better on not normalised data. In the OOD generalization experiment on cMNIST we normalised data also for LMC. Additionally, we performed batch-norm warm before testing in the OOD experiments. In meta-CL (§ F) the batch norms do not keep the estimates of running statistics ($track\_running\_stats$ is set to False) and momentum is set to 1.

# B  Continual Transfer Learning (CTrL)

## B.1  Streams used

As in [92], all input samples were reshaped to a 32 x 32 pixels resolution. We normalized for all methods but LMC unless otherwise stated. We did not use any data augmentation techniques. The datasets used for the first 5 streams are described in the Table 3, datasets used for the streams $S^{long30}$ and $S^{long}$ are described in the Table 4 and Table 5.

| STREAM | | $T_1$ | $T_2$ | $T_3$ | $T_4$ | $T_5$ | $T_6$ |
|---|---|---|---|---|---|---|---|
| $S^+$ | DATASETS | CIFAR-10[49] | MNIST[51] | DTD[17] | F-MNIST[98] | SVHN[69] | CIFAR-10 |
| | #TRAIN SAMPLES | 4000 | 400 | 400 | 400 | 400 | 400 |
| | #VAL. SAMPLES | 2000 | 200 | 200 | 200 | 200 | 200 |
| $S^-$ | DATASETS | CIFAR-10 | MNIST | DTD | F-MNIST | SVHN | CIFAR-10 |
| | #TRAIN SAMPLES | 400 | 400 | 400 | 400 | 400 | 4000 |
| | #VAL. SAMPLES | 200 | 200 | 200 | 200 | 200 | 2000 |
| $S^{in}$ | DATASETS | R-MNIST | CIFAR-10 | DTD | F-MNIST | SVHN | R-MNIST |
| | #TRAIN SAMPLES | 4000 | 400 | 400 | 400 | 400 | 50 |
| | #VAL. SAMPLES | 2000 | 200 | 200 | 200 | 200 | 30 |
| $S^{out}$ | DATASETS | CIFAR-10 | MNIST | DTD | F-MNIST | SVHN | CIFAR-10 |
| | #TRAIN SAMPLES | 4000 | 400 | 400 | 400 | 400 | 400 |
| | #VAL. SAMPLES | 2000 | 200 | 200 | 200 | 200 | 200 |
| $S^{pl}$ | DATASETS | MNIST | DTD | F-MNIST | SVHN | CIFAR-10 | |
| | #TRAIN SAMPLES | 400 | 400 | 400 | 400 | 4000 | |
| | #VAL. SAMPLES | 200 | 200 | 200 | 200 | 2000 | |

Table 3: Details on the datasets and training/validation data amounts for the used streams [92].

## B.2  Baselines and training details

We adopted the original soft-gating with fixed modules (**SG-F**) proposed by Mendez and Eaton [63] in two ways: (1) instead or relying on a pool of initially pretrained modules shared across all layers, we initialize a separate set of modules per layer. This is necessary in order to comply with the experimental setup of CTrL which does not allow pretraining. (2) We used the expansion strategy proposed in MNTDP [92] for SG-F, i.e. for each task different layouts with no or one new module per-layer starting at the top layer are trained, the layout with the best validation accuracy is accepted. The original expansion strategy of [63] is similar in spirit, yet relies on a module pool shared between layers and an initial pretraining of modules, which allows training of only two parallel models: with and without adding a single new module to the shared pool.

In **SG-F**(A) we share a single controller network among all tasks in the sequence. Thereby, the main network still uses task IDs to select the task-specific output head. In the controller a single head architecture is used to gate the modules. As modules are added to the learner, each head of the controller used to gate each layer of the main learner is also expanded. This baseline showcases forgetting in the controller if it is shared across tasks.

For **HAT** we used a separate batch-norm layer for each task. Using shared batch-norm resulted in high forgetting rate for this method.

| Task | Dataset | Classes | # Train | # Val | # Test |
|---|---|---|---|---|---|
| 0 | CIFAR10 | DEER, TRUCK, DOG, CAT, BIRD | 25 | 15 | 5000 |
| 1 | MNIST | MNIST 6 - SIX,0 - ZERO,7 - SEVEN,8 - EIGHT,4 - FOUR | 5000 | 2500 | 4894 |
| 2 | FASHION-MNIST | COAT, BAG, TROUSER, DRESS, T-SHIRT/TOP | 5000 | 2500 | 5000 |
| 3 | SVHN | SVHN 9 - NINE,8 - EIGHT,4 - FOUR,0 - ZERO,6 - SIX | 25 | 15 | 5000 |
| 4 | CIFAR100 | WORM, POSSUM, AQUARIUM FISH, ORCHID, LIZARD | 25 | 15 | 500 |
| 5 | CIFAR10 | FROG, AUTOMOBILE, CAT, TRUCK, DOG | 5000 | 2500 | 5000 |
| 6 | SVHN | SVHN 3 - THREE,1 - ONE,5 - FIVE,4 - FOUR,7 - SEVEN | 25 | 15 | 5000 |
| 7 | MNIST | MNIST 4 - FOUR,5 - FIVE,3 - THREE,2 - TWO,7 - SEVEN | 5000 | 2500 | 4874 |
| 8 | FASHION-MNIST | SNEAKER, SANDAL, ANKLE BOOT, COAT, T-SHIRT/TOP | 25 | 15 | 5000 |
| 9 | FASHION-MNIST | DRESS, COAT, ANKLE BOOT, BAG, TROUSER | 5000 | 2500 | 5000 |
| 10 | SVHN | SVHN 3 - THREE,7 - SEVEN,0 - ZERO,1 - ONE,8 - EIGHT | 25 | 15 | 5000 |
| 11 | CIFAR100 | OTTER, LEOPARD, BEETLE, RAY, BUTTERFLY | 2250 | 1250 | 500 |
| 12 | SVHN | SVHN 6 - SIX,1 - ONE,9 - NINE,2 - TWO,0 - ZERO | 25 | 15 | 5000 |
| 13 | FASHION-MNIST | SNEAKER, ANKLE BOOT, T-SHIRT/TOP, SANDAL, DRESS | 5000 | 2500 | 5000 |
| 14 | MNIST | MNIST 5 - FIVE,1 - ONE,9 - NINE,7 - SEVEN,8 - EIGHT | 5000 | 2500 | 4866 |
| 15 | MNIST | MNIST 5 - FIVE,6 - SIX,7 - SEVEN,9 - NINE,2 - TWO | 25 | 15 | 4850 |
| 16 | SVHN | SVHN 4 - FOUR,0 - ZERO,1 - ONE,2 - TWO,7 - SEVEN | 25 | 15 | 5000 |
| 17 | FASHION-MNIST | T-SHIRT/TOP, SNEAKER, SHIRT, TROUSER, SANDAL | 25 | 15 | 5000 |
| 18 | CIFAR10 | CAT, FROG, BIRD, SHIP, DEER | 5000 | 2500 | 5000 |
| 19 | SVHN | SVHN 9 - NINE,2 - TWO,8 - EIGHT,4 - FOUR,7 - SEVEN | 25 | 15 | 5000 |
| 20 | CIFAR10 | SHIP, HORSE, DOG, TRUCK, CAT | 25 | 15 | 5000 |
| 21 | FASHION-MNIST | SNEAKER, T-SHIRT/TOP, SHIRT, DRESS, PULLOVER | 5000 | 2500 | 5000 |
| 22 | CIFAR10 | AIRPLANE, TRUCK, DEER, FROG, BIRD | 5000 | 2500 | 5000 |
| 23 | SVHN | SVHN 2 - TWO,6 - SIX,4 - FOUR,1 - ONE,5 - FIVE | 5000 | 2500 | 5000 |
| 24 | MNIST | MNIST 8 - EIGHT,3 - THREE,9 - NINE,4 - FOUR,7 - SEVEN | 25 | 15 | 4956 |
| 25 | SVHN | SVHN 4 - FOUR,8 - EIGHT,2 - TWO,6 - SIX,7 - SEVEN | 25 | 15 | 5000 |
| 26 | SVHN | SVHN 1 - ONE,4 - FOUR,7 - SEVEN,9 - NINE,2 - TWO | 25 | 15 | 5000 |
| 27 | CIFAR100 | SWEET PEPPER, COCKROACH, MOTORCYCLE, TANK, ELEPHANT | 25 | 15 | 500 |
| 28 | SVHN | SVHN 3 - THREE,2 - TWO,4 - FOUR,7 - SEVEN,1 - ONE | 5000 | 2500 | 5000 |
| 29 | CIFAR100 | CHIMPANZEE, STREETCAR, WOLF, BEAVER, ROSE | 25 | 15 | 500 |
| 30 | CIFAR10 | HORSE, AIRPLANE, DEER, AUTOMOBILE, TRUCK | 25 | 15 | 5000 |
| 31 | SVHN | SVHN 5 - FIVE,8 - EIGHT,7 - SEVEN,4 - FOUR,3 - THREE | 5000 | 2500 | 5000 |
| 32 | FASHION-MNIST | COAT, DRESS, SANDAL, PULLOVER, T-SHIRT/TOP | 5000 | 2500 | 5000 |
| 33 | CIFAR10 | HORSE, SHIP, TRUCK, FROG, CAT | 25 | 15 | 5000 |
| 34 | CIFAR10 | SHIP, DOG, BIRD, AIRPLANE, CAT | 25 | 15 | 5000 |
| 35 | CIFAR10 | DEER, AIRPLANE, SHIP, TRUCK, AUTOMOBILE | 5000 | 2500 | 5000 |
| 36 | CIFAR100 | BOY, BEAVER, WILLOW TREE, SHARK, TANK | 25 | 15 | 500 |
| 37 | SVHN | SVHN 3 - THREE,4 - FOUR,9 - NINE,1 - ONE,8 - EIGHT | 25 | 15 | 5000 |
| 38 | SVHN | SVHN 9 - NINE,4 - FOUR,5 - FIVE,3 - THREE,1 - ONE | 25 | 15 | 5000 |
| 39 | CIFAR10 | FROG, AIRPLANE, CAT, DOG, TRUCK | 25 | 15 | 5000 |
| 40 | CIFAR10 | SHIP, DEER, TRUCK, HORSE, BIRD | 25 | 15 | 5000 |
| 41 | FASHION-MNIST | DRESS, SHIRT, TROUSER, COAT, SNEAKER | 25 | 15 | 5000 |
| 42 | CIFAR100 | STREETCAR, BEAVER, TIGER, BUS, RACCOON | 25 | 15 | 500 |
| 43 | FASHION-MNIST | COAT, BAG, DRESS, SNEAKER, SANDAL | 25 | 15 | 5000 |
| 44 | MNIST | MNIST 5 - FIVE,9 - NINE,7 - SEVEN,6 - SIX,2 - TWO | 5000 | 2500 | 4850 |
| 45 | CIFAR100 | HAMSTER, PINE TREE, COCKROACH, BOY, COUCH | 25 | 15 | 500 |
| 46 | MNIST | MNIST 0 - ZERO,3 - THREE,2 - TWO,7 - SEVEN,9 - NINE | 5000 | 2500 | 4980 |
| 47 | FASHION-MNIST | SANDAL, DRESS, COAT, TROUSER, BAG | 25 | 15 | 5000 |
| 48 | SVHN | SVHN 0 - ZERO,8 - EIGHT,5 - FIVE,2 - TWO,1 - ONE | 5000 | 2500 | 5000 |
| 49 | CIFAR10 | HORSE, FROG, AIRPLANE, DOG, SHIP | 5000 | 2500 | 5000 |

Table 4: Details on the datasets and training/validation data amounts used for $S^{long}$ (Part 1) [92].

For all task ID aware methods, the task ID was used either to select the task specific output head (as in HAT, EWC or ER) or the task specific structure as in MNTDP (which includes the output head). Thereby, we treat first and last tasks in $S^+$, $S^-$, $S^{in}$ and $S^{out}$ streams as tasks with different IDs. This corresponds to the definition provided in [92], where the task ID is defined to correspond to the sequential order of the task in the sequence.

We used Adam optimizer for all baseline methods but HAT [86]. Using Adam for HAT resulted in more forgetting, which we believe is because HAT masks out only the gradients of some parameters and does not effect Adam's momentum.

**Hyper-parameter and model selection** was performed using average mean validation accuracy over all tasks in the stream (stream level) with splits detailed in Table 3. When varying the seeds in the provided experiments, we did not very the seed that effects data generation (CTrL Streams) but only the seed that affected the algorithm, model initialization as well as data-loader's batch sampling.

## B.3 Metrics

Here we formally define metrics used in the experiments. These metrics are similar to the ones used by Veniat et al. [92]. $\Delta$ denotes the prediction accuracy of the predictor $\mathcal{F}$. We use subscripts to indicate the version of the parameters: e.g. $\theta_{1...t}$ indicates the functional parameters of the learner after it was continually trained on $t$ tasks, while $\theta_T$ indicates the version of functional parameters after learning only task $T$ in isolation.

| TASK | DATASET | CLASSES | # TRAIN | # VAL | # TEST |
|------|---------|---------|---------|-------|--------|
| 50 | SVHN | SVHN 9 - NINE,4 - FOUR,6 - SIX,5 - FIVE,2 - TWO | 25 | 15 | 5000 |
| 51 | SVHN | SVHN 3 - THREE,6 - SIX,8 - EIGHT,9 - NINE,1 - ONE | 25 | 15 | 5000 |
| 52 | CIFAR100 | CROCODILE, LION, BUTTERFLY, OTTER, HAMSTER | 2250 | 1250 | 500 |
| 53 | MNIST | MNIST 9 - NINE,8 - EIGHT,6 - SIX,7 - SEVEN,3 - THREE | 25 | 15 | 4932 |
| 54 | MNIST | MNIST 7 - SEVEN,3 - THREE,8 - EIGHT,4 - FOUR,2 - TWO | 25 | 15 | 4956 |
| 55 | SVHN | SVHN 4 - FOUR,2 - TWO,6 - SIX,0 - ZERO,5 - FIVE | 25 | 15 | 5000 |
| 56 | CIFAR100 | SEA, CHAIR, SNAKE, SPIDER, SNAIL | 25 | 15 | 500 |
| 57 | CIFAR100 | BEETLE, TELEVISION, TABLE, PORCUPINE, CUP | 25 | 15 | 500 |
| 58 | CIFAR10 | CAT, HORSE, FROG, TRUCK, AUTOMOBILE | 25 | 15 | 5000 |
| 59 | SVHN | SVHN 8 - EIGHT,6 - SIX,1 - ONE,5 - FIVE,3 - THREE | 25 | 15 | 5000 |
| 60 | CIFAR10 | BIRD, FROG, HORSE, SHIP, DEER | 25 | 15 | 5000 |
| 61 | MNIST | MNIST 1 - ONE,9 - NINE,8 - EIGHT,7 - SEVEN,2 - TWO | 25 | 15 | 4974 |
| 62 | FASHION-MNIST | DRESS, T-SHIRT/TOP, SANDAL, TROUSER, SNEAKER | 25 | 15 | 5000 |
| 63 | MNIST | MNIST 6 - SIX,4 - FOUR,0 - ZERO,7 - SEVEN,8 - EIGHT | 25 | 15 | 4894 |
| 64 | SVHN | SVHN 4 - FOUR,2 - TWO,7 - SEVEN,6 - SIX,3 - THREE | 5000 | 2500 | 5000 |
| 65 | CIFAR100 | PEAR, SKYSCRAPER, SHARK, PLAIN, DOLPHIN | 2250 | 1250 | 500 |
| 66 | CIFAR10 | FROG, BIRD, AIRPLANE, SHIP, HORSE | 25 | 15 | 5000 |
| 67 | CIFAR10 | FROG, DEER, SHIP, HORSE, TRUCK | 25 | 15 | 5000 |
| 68 | CIFAR10 | HORSE, DEER, TRUCK, AIRPLANE, DOG | 25 | 15 | 5000 |
| 69 | CIFAR100 | SKUNK, ORCHID, CATTLE, SPIDER, LOBSTER | 25 | 15 | 500 |
| 70 | MNIST | MNIST 3 - THREE,5 - FIVE,4 - FOUR,9 - NINE,1 - ONE | 25 | 15 | 4874 |
| 71 | SVHN | SVHN 4 - FOUR,3 - THREE,1 - ONE,7 - SEVEN,5 - FIVE | 25 | 15 | 5000 |
| 72 | FASHION-MNIST | COAT, DRESS, BAG, SANDAL, TROUSER | 25 | 15 | 5000 |
| 73 | FASHION-MNIST | SANDAL, DRESS, ANKLE BOOT, PULLOVER, SHIRT | 25 | 15 | 5000 |
| 74 | MNIST | MNIST 3 - THREE,2 - TWO,8 - EIGHT,6 - SIX,4 - FOUR | 25 | 15 | 4914 |
| 75 | CIFAR10 | AIRPLANE, DOG, HORSE, BIRD, SHIP | 25 | 15 | 5000 |
| 76 | CIFAR10 | AUTOMOBILE, HORSE, AIRPLANE, CAT, TRUCK | 25 | 15 | 5000 |
| 77 | FASHION-MNIST | SANDAL, COAT, SHIRT, DRESS, ANKLE BOOT | 25 | 15 | 5000 |
| 78 | FASHION-MNIST | TROUSER, T-SHIRT/TOP, SANDAL, SNEAKER, DRESS | 25 | 15 | 5000 |
| 79 | CIFAR100 | LION, TURTLE, CUP, SHREW, ROSE | 25 | 15 | 500 |
| 80 | MNIST | MNIST 2 - TWO,4 - FOUR,5 - FIVE,6 - SIX,1 - ONE | 25 | 15 | 4832 |
| 81 | CIFAR100 | TURTLE, MOUNTAIN, KANGAROO, LOBSTER, CRAB | 25 | 15 | 500 |
| 82 | FASHION-MNIST | SANDAL, SNEAKER, T-SHIRT/TOP, COAT, PULLOVER | 25 | 15 | 5000 |
| 83 | CIFAR100 | PLAIN, SKYSCRAPER, BUTTERFLY, TRAIN, SEA | 25 | 15 | 500 |
| 84 | MNIST | MNIST 9 - NINE,5 - FIVE,4 - FOUR,8 - EIGHT,2 - TWO | 25 | 15 | 4848 |
| 85 | SVHN | SVHN 1 - ONE,7 - SEVEN,0 - ZERO,5 - FIVE,6 - SIX | 25 | 15 | 5000 |
| 86 | MNIST | MNIST 2 - TWO,4 - FOUR,7 - SEVEN,3 - THREE,8 - EIGHT | 25 | 15 | 4956 |
| 87 | CIFAR10 | SHIP, AUTOMOBILE, FROG, DOG, HORSE | 25 | 15 | 5000 |
| 88 | CIFAR100 | CLOUD, SPIDER, TIGER, MOUSE, SNAKE | 25 | 15 | 500 |
| 89 | FASHION-MNIST | DRESS, PULLOVER, T-SHIRT/TOP, BAG, SHIRT | 25 | 15 | 5000 |
| 90 | CIFAR10 | AUTOMOBILE, TRUCK, CAT, DOG, HORSE | 25 | 15 | 5000 |
| 91 | MNIST | MNIST 0 - ZERO,8 - EIGHT,9 - NINE,7 - SEVEN,5 - FIVE | 25 | 15 | 4846 |
| 92 | MNIST | MNIST 3 - THREE,9 - NINE,7 - SEVEN,5 - FIVE,8 - EIGHT | 25 | 15 | 4866 |
| 93 | FASHION-MNIST | BAG, COAT, T-SHIRT/TOP, ANKLE BOOT, TROUSER | 25 | 15 | 5000 |
| 94 | CIFAR100 | CAMEL, TRACTOR, ORCHID, PEAR, AQUARIUM FISH | 25 | 15 | 500 |
| 95 | MNIST | MNIST 2 - TWO,8 - EIGHT,9 - NINE,4 - FOUR,3 - THREE | 25 | 15 | 4956 |
| 96 | MNIST | MNIST 9 - NINE,8 - EIGHT,4 - FOUR,0 - ZERO,7 - SEVEN | 25 | 15 | 4936 |
| 97 | FASHION-MNIST | BAG, DRESS, SHIRT, SANDAL, PULLOVER | 25 | 15 | 5000 |
| 98 | CIFAR100 | MOUSE, SNAIL, BED, TROUT, GIRL | 25 | 15 | 500 |
| 99 | FASHION-MNIST | TROUSER, PULLOVER, SANDAL, T-SHIRT/TOP, ANKLE BOOT | 25 | 15 | 5000 |

Table 5: Details on the datasets and training/validation data amounts used for $S^{long}$ (Part 2).

**Average accuracy** on all tasks seen so far.

$$\mathcal{A} = \frac{1}{T} \sum_{t=1}^{T} \mathbb{E}_{(x,y) \sim \mathcal{D}_t} [\Delta\Big(\mathcal{F}(x; \theta_{1...T}, \phi_{1...T}), y\Big)] \tag{14}$$

**Forgetting** — the average loss of accuracy on a task at the end of training as compared to the first time the task was seen. Positive values indicate positive backward transfer.

$$\mathcal{F} = \frac{1}{T-1} \sum_{t=1}^{T} \mathbb{E}_{(x,y) \sim \mathcal{D}_t} [\Delta\Big(\mathcal{F}(x; \theta_{1...T}, \phi_{1...T}), y\Big) - \Delta\Big(\mathcal{F}(x; \theta_{1...t}, \phi_{1...t}), y\Big)] \tag{15}$$

**Transfer** — the difference in performance on the last ($T$'th) task between the modular learner trained on the entire sequence and an expert $\mathcal{F}'$ trained on the last task in isolation.

$$\mathcal{T} = \mathbb{E}_{(x,y) \sim \mathcal{D}_T} \Delta\Big(\mathcal{F}(x; \theta_{1...T}, \phi_{1...T}), y\Big) - \Delta\Big(\mathcal{F}'(x; \theta_T), y\Big) \tag{16}$$

### B.4 Transfer results CTrL

| MODEL | $S^-$ | | | $S^+$ | | | $S^{in}$ | | | $S^{out}$ | | |
|---|---|---|---|---|---|---|---|---|---|---|---|---|
| | Acc $t_1$ | Acc $t_1^-$ | $\mathcal{T}$ | Acc $t_1$ | Acc $t_1^+$ | $\mathcal{T}$ | Acc $t_1$ | Acc $t_1'$ | $\mathcal{T}$ | Acc $t_1$ | Acc $t_1''$ | $\mathcal{T}$ |
| EXPERTS | 65.5±0.7 | 41.8±1.0 | 0 | 41.3±2.9 | 65.6±0.5 | 0 | 98.5±0.2 | 76.9±4.9 | 0 | 65.9±0.6 | 43.5±1.6 | 0 |
| MNTDP | 63.0±3.6 | 56.9±5.1 | 15.1 | 43.2±0.7 | 65.9±0.8 | 0.3 | 98.9±0.1 | 93.3±1.6 | 16.4 | 65.0±1.2 | 57.7±1.7 | 14.2 |
| LMC(¬A) | 65.2±0.4 | 60.0±1.1 | 18.2 | 42.9±0.9 | 60.6±1.9 | -4.7 | 98.7±0.1 | 92.5±7.6 | 15.6 | 65.2±0.2 | 59.8±1.1 | 16.3 |
| LMC(A) | 65.2±0.4 | 63.0±1.7 | 21.2 | 43.1±0.6 | 62.2±0.7 | -3.4 | 98.7±0.1 | 88.3±1.6 | 11.4 | 65.5±0.6 | 42.0±21.9 | -1.5 |
| S.G+FX | 64.9±0.4 | 49.1±7.3 | 7.3 | 43.1±0.4 | 61.7±1.7 | -3.9 | 98.8±0.1 | 80.4±6.8 | 3.5 | 65.0±0.4 | 51.5±6.5 | 8.0 |

Table 6: Transfer results on the CTrL benchmark. We provide the accuracy of the first and the last task on each of the streams. Additionally, we measure transfer $\mathcal{T}$ as the difference between the last task's accuracy of the model trained on the corresponding stream and an expert model trained on the last task in isolation.

### B.5 Module selection: CTrL

In Figure 6 we plot the average module selection of LMC for different streams of the CTrL [92] benchmark after continual training on the corresponding stream. The plots correspond to the runs with the best validation accuracy on the corresponding stream. We observe that for the $S^-$ stream LMC reuses modules trained on the task 1 for the sixth task. On $S^+$, when tested on task 1 and task 6 same modules are used, yet the two modules at the last layer correspond to the ones added when learning task 6 (this task contains more training data than task 1). On $S^{in}$ the last task reuses only the last layer's module from task 1 and several other modules on the previous layers from previous tasks. Finally, for $S^{pl}$ stream, even though the tasks are unrelated, several modules are reused across the tasks. Importantly, since LMC implements soft module selections strategy several modules can be used together for the same batch of samples (see Eq. 6).

### B.6 Ablation of threshold $z'$

To see how the z threshold influences the final number of modules (M) and the average (test) accuracy over seen tasks ($\mathcal{A}$) we report the $\mathcal{A}$ and M for LMC(¬A) on 4 streams with fixed hyperparameters while only varying the threshold $z'$. In Figure 7a we plot the values of the threshold $z'$ on the x-axis against average accuracy $\mathcal{A}$ on the y-axis: higher $z'$ leads to fewer modules being instantiated resulting in lower average accuracy $\mathcal{A}$. Additionally, in Figure 7b we plot the same runs but now with number of modules on the x-axis and the average accuracy $\mathcal{A}$ on the y-axis. We identify the number of modules and accuracy of the MNTDP baseline with dotted lines with the corresponding stream colors in both plots. In Figure 7b we observe that LMC instantiates a comparable number of modules as MNTDP in order to reach similar accuracy, with an exception of the $S^{pl.}$ stream, where LMC tends to add mode modules to reach a similar accuracy.

## C   Long sequences hyperparameter search visualization

We plot number of modules against the average test accuracy over all seen tasks ($\mathcal{A}$) in Figure 8 for all executed hyperparameter search runs on both $S^{long30}$ and $S^{long}$ sequences.

## D   Combining independent models

The aim of this experiment is to show the ability of independently trained LMC models to be combined without fine-tuning, without loss in performance and also enabling positive transfer. In this experiment the test set's distributions of the cMNIST tasks are different from the training/validation sets' distributions (see § 4.3). We used oracle model selection choosing hyperparameters on the test set. Note, that model selection is an unsolved challenge in the OOD generalization literature, where selecting model using oracle strategies is sometimes excepted if the baselines methods are also tuned using oracle strategies [30, 77].

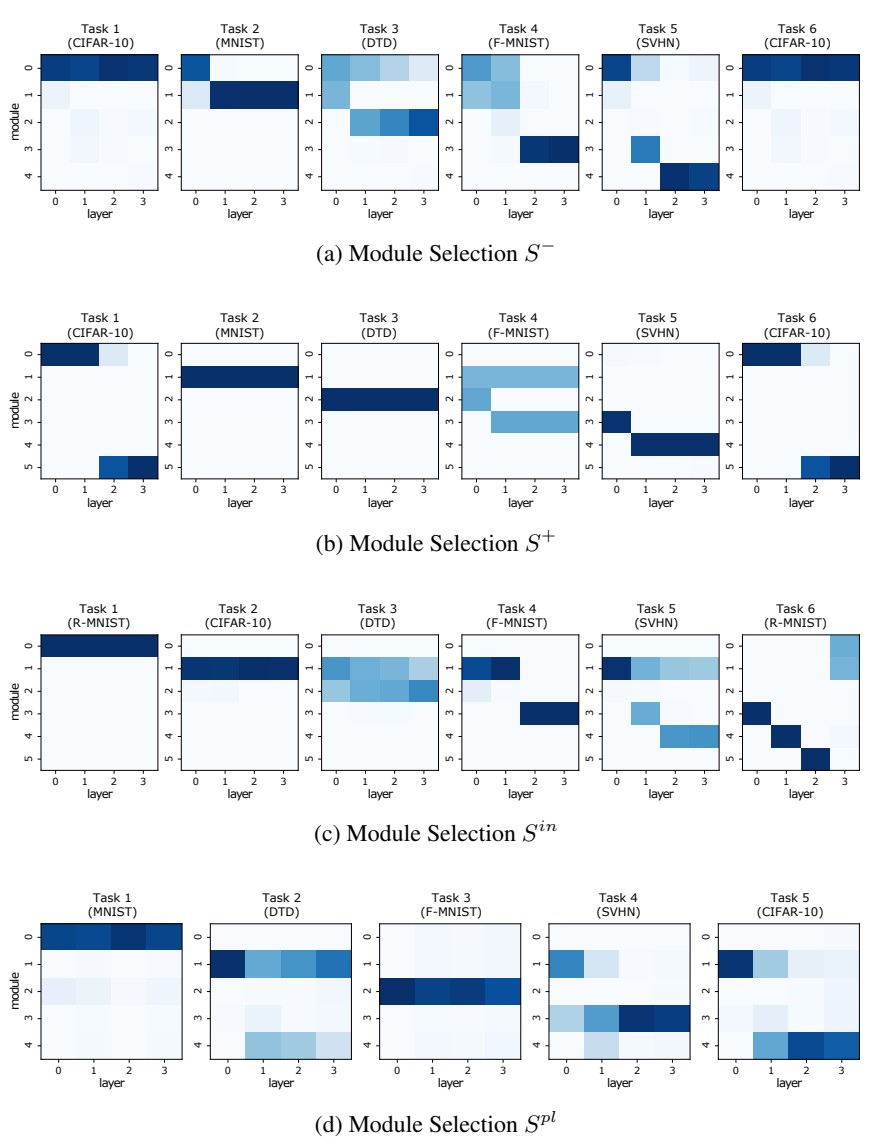

(a) Module Selection $S^-$

(b) Module Selection $S^+$

(c) Module Selection $S^{in}$

(d) Module Selection $S^{pl}$

Figure 6: Average module selection for different streams at test time. Horizontal axis corresponds to the layer number, vertical axis corresponds to the module index at a layer; darker color corresponds to higher average activation strength (averaged over batches in the corresponding task).

# E   Compositional OOD generalization

Some examples of the raw samples used for this experiment are presented in Figure 10. Model selection was performed assuming access to the test sets of the OOD tasks. As mentioned in § D, this is sometimes an excepted strategy is settings in which test distribution is different from the training/validation distribution. Additionally, the input samples were normalized using statistics computed on training set of each tasks, including the OOD tasks (for which no training was performed).

In the main paper we presented a version of LMC with **omitted projection phase**. More precisely this means that the structural loss of the modules above was not propagated into the free modules on the lower layers (closer to the input) and that module addition was allowed during the whole training process.

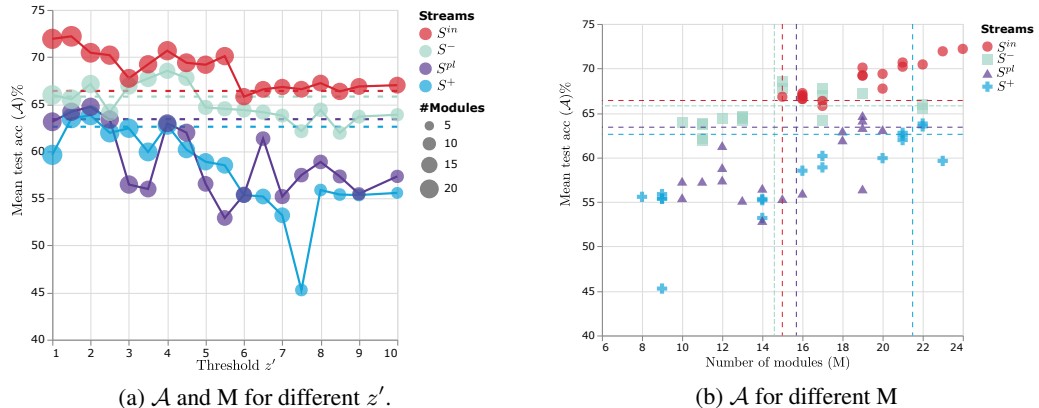

(a) $\mathcal{A}$ and M for different $z'$.        (b) $\mathcal{A}$ for different M

Figure 7: (a) Mean test accuracy $\mathcal{A}$ and number of modules M for different values of the threshold hyperparameter $z'$ for LMC$_{(\neg A)}$. Dotted lines mark the accuracy of MNTDP[92]. (b) Number of modules (x-axis) against average accuracy $\mathcal{A}$ for runs with different threshold $z'$. Dotted lines mark $\mathcal{A}$ and M for MNTDP[92] (best seen in color).

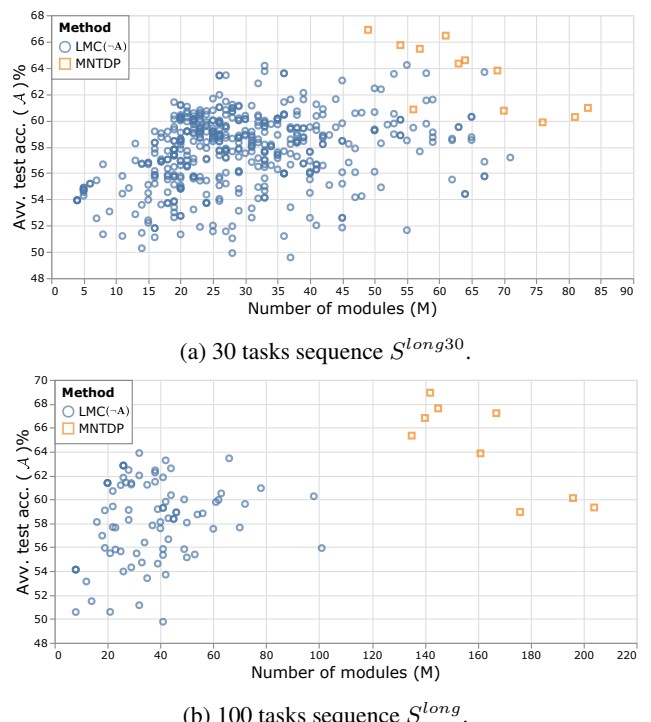

(a) 30 tasks sequence $S^{long30}$.

(b) 100 tasks sequence $S^{long}$.

Figure 8: Results on $S^{long}$ and $S^{long30}$ sequences for different hyperparameter values for all executed hyperparameter search runs (this is an expanded version of Figure 5).

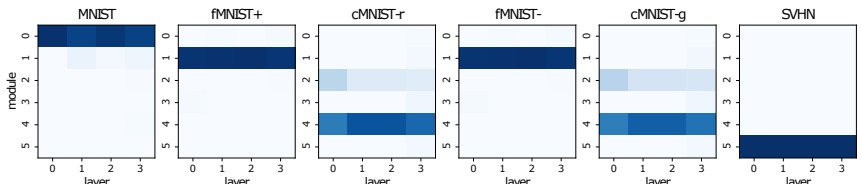

Figure 9: Average module selection per task on LMC3, which was constructed from plugging together independently trained LMC1 (first 3 tasks) and LMC2 (last 3 tasks).

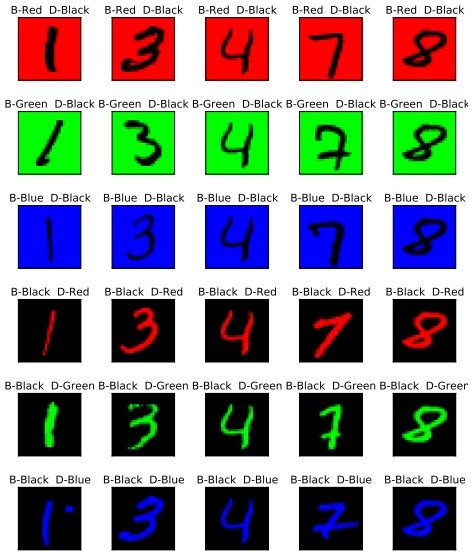

Figure 10: Examples of samples from the used colored-MNIST dataset.

## E.1  Module selection

In Figure 11 we plot average module selection per task for the compositional OOD generalization setting.

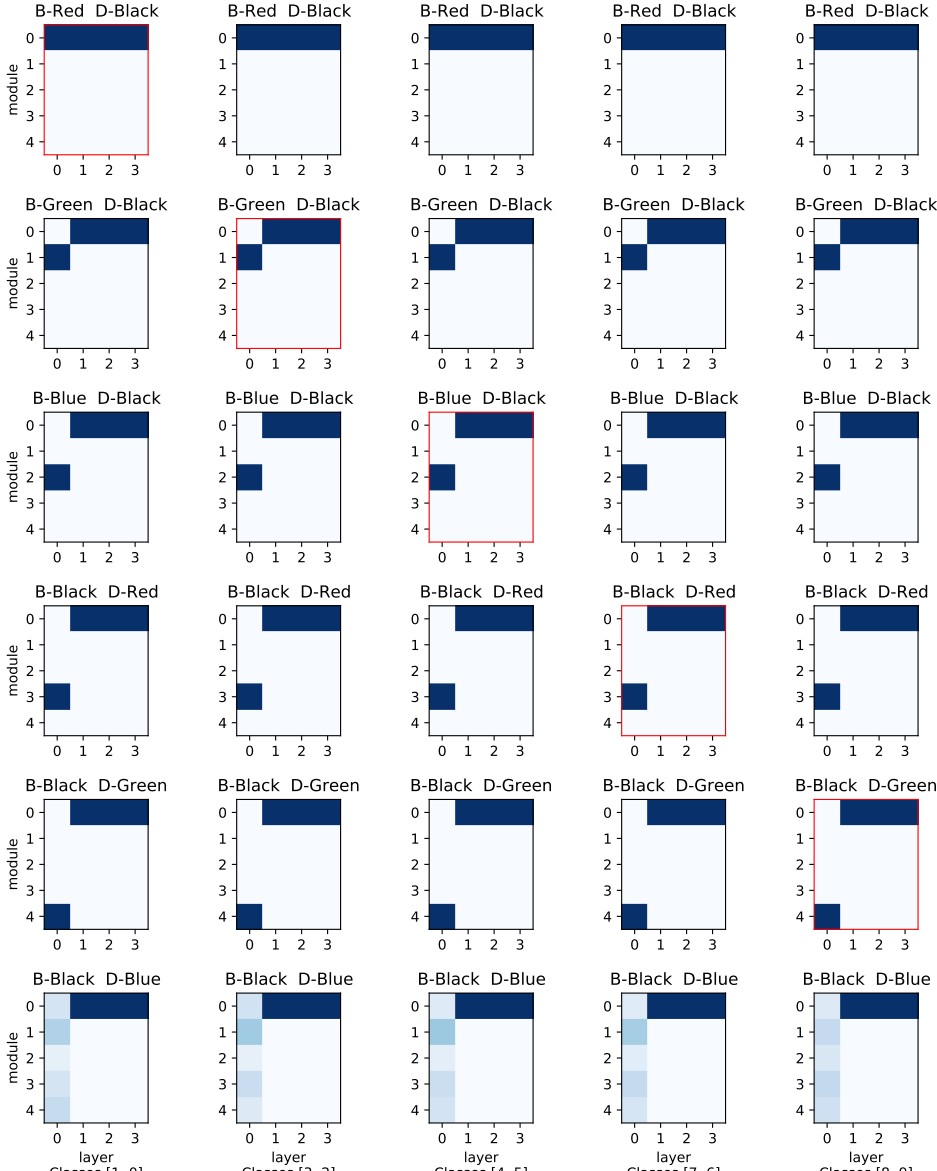

Figure 11: Average module selection per task for compositional OOD generalization setting presented in § 4.2. Each column represents a task (e.g. 0vs.1 classification), each row corresponds to a digit-background color combination. Each square represents a modular learner, x-axis is the layer, y-axis is the modules at each layer. The color intensities correspond to the average activation strength of the module when tested on the corresponding task (the darker the higher). **B** - stands for Background and **D** - stands for digit (e.g. "B-Blue D-Black" stands for blue background and black digit). The diagonal tasks (with red boarder) are seen during the continual training phase, after which the model is evaluated on all 5 combinations of seen digit-background colors for each task. The last row corresponds to the module selection for tasks with digit-background color combination which was not seen during the continual learning phase (black background+blue digit).

# F  Continual meta-learning

In the continual meta-learning setting, a learner is exposed to tasks sampled from a sequence of environments. The goal of this setting is to construct a learner that quickly (i.e. within a few steps of gradient descent) learns tasks from new environments and relearns (or remembers) tasks from

previously learned environments [41, 14, 34]. Methods applicable in this setting usually rely on gradient-based meta-learning strategies based on MAML [24]. The goal of MAML is to learn a good parameter initialization $\theta_0$ such that the learner can achieve low loss $\mathcal{L}_\tau$ on a randomly sampled task $\tau$ after a few steps of gradient descent. The objective of MAML can be formulated as [71]:

$$\theta_0 = arg \min_\theta \mathbb{E}_\tau(\mathcal{L}_\tau(U_\tau^k(\theta))), \tag{17}$$

where where $U$ is the operator that performs $k$ inner-loop SGD updates starting with $\theta$ using data samples from task $\tau$. MAML has been extensively used in the few-shot learning scenario [94], where similar tasks are sampled from a stationary distribution. However, as pointed out by Jerfel et al. [41], it is unrealistic to assume the existence of a single set of meta-parameters that is close to all tasks in heterogeneous settings with outliers and non-stationary task distributions. In such setting, dissimilar tasks worsen generalization and non-stationarity causes catastrophic forgetting [46]. Instead of learning a single set of meta-parameters for all the tasks, Jerfel et al. [41] propose to learn a set of monolithic expert models, each representing a separate parameter initialization. The meta-learner then leverages the connection between gradient-based meta-learning and hierarchical Bayes [29] for selecting the most likely expert for the given task at hand. The probability of an expert to be selected is calculated based on the inner-loss of each expert as well as a non-parameteric Dirichlet process prior estimated in an online fashion.

Here, we construct two environment sequences. In the first one, the learner is meta-trained on a sequence of Omniglot [50], MNIST [51] and fashion-MNIST [98] datasets — the OMF sequence. We evaluate models in two settings. In the first setting, we report the average meta-test test accuracy over the environments seen so far. In the second setting, models are evaluated on environment mixtures, i.e. each test task is composed of samples from the environments learned so far. A possible mixture task could contain classes 1 and 2 sampled from Omniglot, 2 and 3 from MNIST and class 5 sampled from the fashion-MNIST environment. The second sequence of environments is constructed from classes of the *mini*Imagenet [79].

We compare to the following baselines. **MAML**: non-modular MAML [24]. **MAML + ER**: MAML with experience replay — we maintain a memory buffer of tasks sampled from previously seen environments, using a reservoir sampling procedure we retrieve a set of previously seen tasks which we then add to the task batches used for meta-training on new environments; **Meta-gating** tries to learn the relevance scores $\gamma^{(l)}$, used for gating, directly in the inner-loop. This is similar to soft-layer ordering used in [63]; **Experts**: an upper bound where separate monolithic model is trained for each environment and selected using the environment ID. We test LMCs with different levels of modular granularity: **LMCe** refers to a model that trains separate experts, but expert selection is performed using an expert-level structural component; **LMCm** is the modular LMC with module selection performed at each layer (as described in § 3). In our experiments we meta-learn the structural components: in the inner-loop we update functional parameters $\theta_m$ of all modules (also the fixed ones) whereas the structural parameters $\phi_m$ are kept constant, in the outer-loop both functional and structural parameters of the free modules are updated. *Importantly, we do not apply the projection phase in the meta-learning experiments.* All experiments are performed in a 5-way 5-shot setting.

For LMCm we used encoder-decoder architectures in the *mini*Imagenet experiments (§ F.3), while for the OMF sequence the invertible structural component was used for the entire network. Note, that task inference is done automatically using the provided meta-test train (query) data, hence, there is no need to use multiple classification heads. For the OMF task sequence each module consists of a single convolutional layer with 64 3x3 filters (padding 1), batch-normalization, ReLU activation function and a max-pooling layer with the kernel-size of 2. This resembles the architecture introduced by Finn et al. [24]. One epoch of meta-training consists of 100 meta-updates, each performed on a batch of 25 tasks (5-way, 5-shot). For the *mini*ImageNet experiment, a single convolutional layer contains 32 filters. One epoch of meta-training performs 100 meta-updates, each on a batch of 4 tasks (5-way 5-shot regime). All learners contain 4 layers, non-modular (MAML, MAML+ER) learners do not expand. In the "Experts" baseline, each expert network corresponds to a 4-layered net with a single module per layer.

Model selection was performed using average meta-test validation accuracy over all environments. We randomly selected 10% of the train datasets for validation purposes for MNIST and fMNIST datasets. For these datasets we did not evaluate the meta-generalization ability of the model, since the classes in meta-train and meta-test splits are the same. For the Omniglot dataset we selected 100 classes for validation purposes. We also flipped the background color of the Omniglot dataset to be

black, which corresponds to the background colors of the MNIST and fMNIST datasets. For the *mini*ImageNet dataset we split the 100 classes in 64-train/16-validation/20-test as in [79].

## F.1 Module addition continual meta-learning.

In this section we explain some details about the adaptation of LMC's expansion strategy (§3.1) for continual meta-learning.

The decision about module addition can be made per sample, per task, or per batch of tasks. In the per sample case, a new module is added whenever a sample is regarded as an outlier by all modules at a layer. In the per task case, a new module is added whenever the average z-score for a task is larger than a threshold. In the per batch of tasks case, the average z-score is calculated over a batch of tasks. In our experiments we found that adding new modules on a per batch of tasks base yields the best efficacy in the continual meta-learning setting.

We do not freeze the modules at the environment switch, but create a checkpoint of each non-frozen (free) module. Thus, modules are allowed to learn in the outer loop until the module addition is triggered. Whenever the module addition is triggered at a layer, the existing free module is dropped back to its state from the most recent checkpoint (i.e. its state at last environment switch).

## F.2 OMF results

Results on OMF are shown in Figure 12. These results suggest that LMC can successfully avoid catastrophic forgetting achieving final average accuracy comparable to the experience replay (MAML + ER) baseline. This can be mainly attributed to the fact that learned modules at a layer are frozen every time a new module is added — modules are not updated on new tasks if these tasks have triggered module addition and thus were recognized as outliers at a given layer.

Results with mixtures of tasks are in Figure 12b. We find that LMC attains better performance than the other baselines. The performance improvement is due to LMC assigning modules on a per-sample basis in each forward pass, effectively guiding the inner- and outer-loop gradients of the meta-training procedure to the modules with higher activation on the current input. This contrasts the selection mechanism proposed by [41] for mixture-of-experts, where estimating the probability of each expert involves computation of the inner-loss which can be performed only on a per-task level.

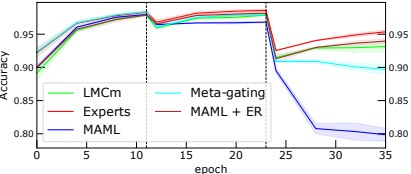

(a) Average meta-test accuracy so far

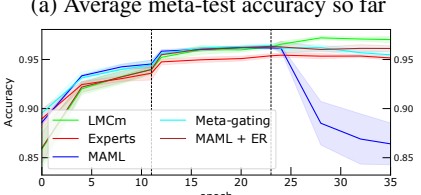

(b) Mixtures of tasks meta-test accuracy

Figure 12: Results on the OMF sequence. Vertical dashed lines mark the environment switches.

While replay slightly underperforms LMC on the mixture task, MAML is unable to reach high accuracy due to catastrophic forgetting. Regarding the experts baseline, each expert only specializes on one of the three environments, which results in low performance on the mixture-task.

In Figure 13 we show how the meta-test test accuracies evolve over the course of continual meta-training for each environment of the OMF sequence. Additionally, in Figure 14 we plot the average module selection at meta-test test time on the OMF sequence after the entire sequence has been learned.

## F.3 Additional *mini*Imagenet results.

In this section we present additional results on the evolving *mini*Imagenet sequence where, similarly to [41], each environment is obtained through application of filters 'blur', 'pencil' and 'night'. Figure 16 plots per environment accuracy of each method that results in the average accuracy over the environments seen so far depicted in Figure 15 (a).

Additionally, we design a mixture-task baseline as follows: we select randomly 5 classes from the *mini*ImageNet dataset, sample datapoints for these 5 classes from each of the environments seen so

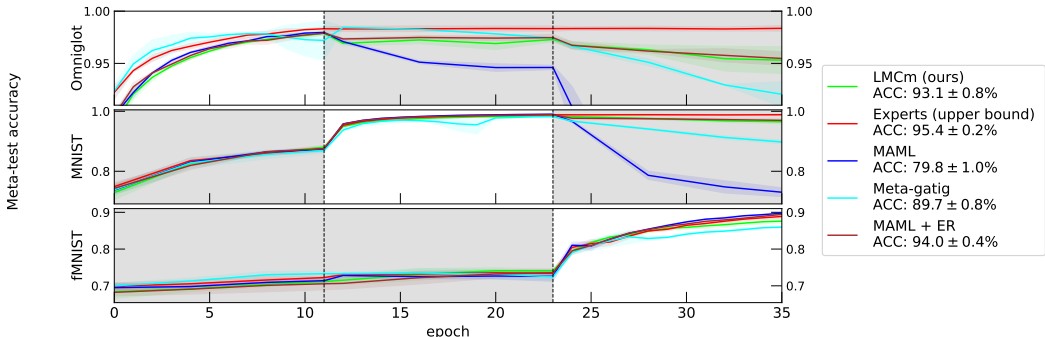

Figure 13: Meta-test test accuracy on the OMF task sequence. Each row shows accuracy on a specific environment, the training time on each environment is highlighted with white background. We also report the average accuracy over all environments at the end of training in the legend (ACC). For this experiment the modular learner (LMCm) was trained using inevitable network as structural component (5-way 5-shot).

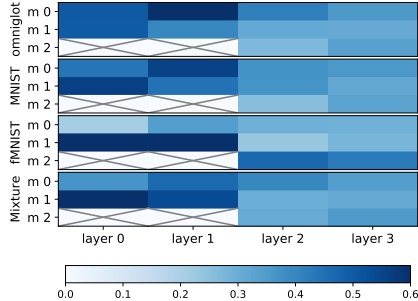

Figure 14: Average module selection of LMCm for a selected run on the OMF task sequence (after continual meta-CL training has been complete on the entire sequence).

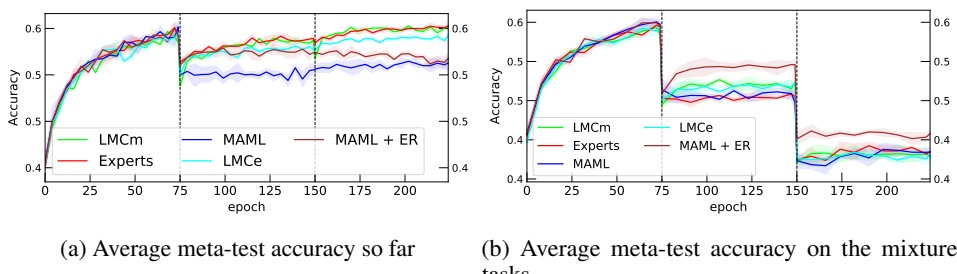

(a) Average meta-test accuracy so far

(b) Average meta-test accuracy on the mixture tasks

Figure 15: Meta-test test results on the sequence of *mini*Imagenet environments. Vertical lines mark environment switches.

far (i.e. all three at the end of the sequence) to build tasks. Hence, a single mixture task for each class will contain samples from each environment seen so far. The learner is then meta-tested on these tasks. This is different from the mixture task designed for the OMF sequence, where classes from different environments were mixed. We present the results on the mixture task in Figure 15(b). We observe that MAML+ER baselines outperforms other learners in this evaluation setting. We hypothesize that poor performance of LMC in this setting is due to the batched modularity procedure (§A.1), where biasing module activation towards majority activation withing the batch of may harm performance, since each batch contains samples from all environments seen so far.

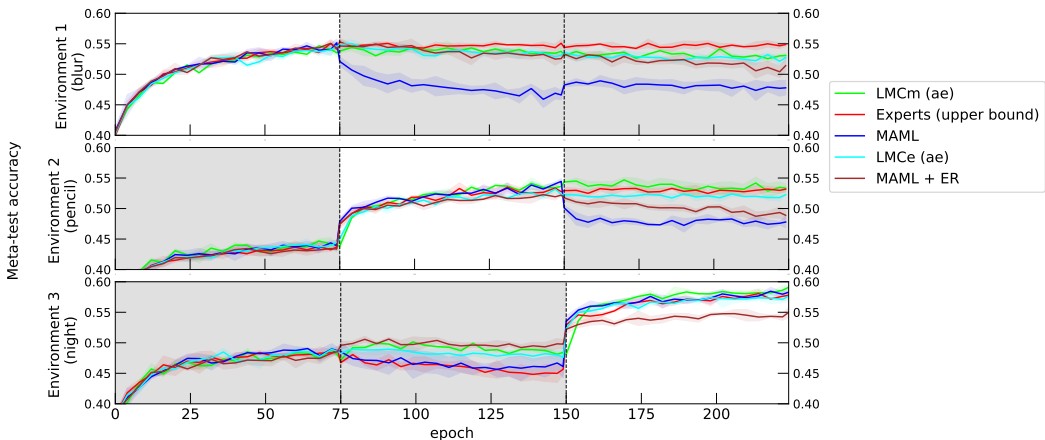

Figure 16: Meta-test test accuracy on the evolving *mini*ImageNet task sequence. Each row shows accuracy on a specific environment, the training time on each environment is highlighted with white background. For this experiment the modular learners (LMCm and LMCe) were trained using decoder as structural component tasked to reconstruct modules' inputs (5-way 5-shot).

# G   Limitations

We identify three main limitations of LMC. The first one is that its computational footprint is not constant with respect to the number of tasks. This is caused by the fact that each forward path through the modular learner requires a forward pass through each module in order to obtain the local relevance scores. Potentially, this can be overcome by assuming local stationarity and using the same modules for several consecutive batches of samples. On the other hand, this might not be a big problem at all: we think of each module as an independent entity that can be executed on it's own dedicated device, in which case at each layer each module's computations can be executed in parallel. The recent trend towards shifting from monolithic to modular architectures can accelerate the development of such hardware devices.

Another limitation of LMC is the lack of direct communication between modules at the same layer, which was shown to be important in some situations such as when different physical processes interact [27, 28]. While explicitly modeling such communication is an interesting direction for the future work, the consolidation of modules through weighed sum in Eq. 4 can be though of as a form of implicit communication, i.e. a form of a shared workspace similar to the one presented in [28]. It is however questionable whether cross-module communication is of any benefit in a standard supervised-learning settings.

Additionally, as discussed in § 4.4, module selection becomes challenging for LMC in presence of large number of candidate modules resulting in lower accuracy on long task sequences as compared to oracle based sleection strategy implemented in MNTDP [92].

Finally, in its working LMC relies on the local OOD detector as well as a generative model as it's structural component. Several recent works have shown that deep generative models often mistakenly assign high likelihood values to outlier points [68, 36, 95]. Hence, LMC's success depends on overcoming these issues through innovation in the fields of OOD detection and generative modeling.

# H   Broader societal impact

This work aimed at leveraging modularity and compositionality for continual learning (CL). The goal of CL is to design systems capable to retain knowledge and transfer knowledge across tasks. Such systems can **positively impact** society in the following ways: (i) models able to retain knowledge withing neural connections do not require storing raw samples in a replay buffer, yielding systems that are more compliant with data privacy standards. (ii) Positive transfer of knowledge across tasks can result in more resource efficient training. (iii) Building modular systems can further improve resource

efficiency: e.g. as shown in the experiment in § 4.3 several modular systems can be combined in a third system without any retraining.

We do not identify any potential **negative societal impacts** of this work in particular beyond the potential negative societal impacts of artificial learning systems in general, which include the risk of decision bias, loss of certain jobs due to automation, risk of increased vulnerability to hacker attacks to name a few.