# OpenReview forum: "Continual Learning via Local Module Composition"
_NeurIPS.cc/2021/Conference — NeurIPS 2021 Poster_

### Official Review · Reviewer_eYQc · 2021-07-08

**Rating:** 6
**Confidence:** 4

**Summary:**

Authors propose an algorithm to perform continual learning without task-id labels at inference. The method involves local module-level OOD estimators which then output linear coefficients that informs the model how to combine outputs. The model also has the option of creating new modules which are first trained in a projection phase from downstream layer signals before starting training on their own. The proposed method is tested in standard continual learning benchmarks as well as OOD detection and meta-learning settings.


**Limitations And Societal Impact:**

Yes.

**Main Review:**

Overall: The proposed method is simple and combines technologies in a reasonable way to achieve good results on a number of settings. I appreciated the breadth of experiments and various supplementary experiments (OOD detection, module coefficient visualizations in the appendix, etc.). I think that the formulation with the two stages of training and the projection phase is fairly elegant. I do have questions about the method and think a few details are inconsistent, but overall enjoyed the paper and feel it’s of sufficient quality to publish at NeurIPS.

Here are a few comments that I would appreciate some feedback on in the author response:

(1) One of the main contributions the paper claims to have is to not require the task ID at testing time. But task-agnostic continual learning is not a new topic, and the authors themselves cited works in this area (e.g. [28] and [30]). So is the main contribution of the paper just the combination of the task-agnosticity with the particular formulation of the module-level training? I’d like to understand in a more concise manner the core impact that the work has from a fundamental science perspective.

(2) Why do the authors not compare to other established task-agnostic baselines? The comparisons are mostly to heavily modified versions of task-id-required methods, and these modifications are not too convincing (for example, using MNTDP with entropy is a bit dicey as entropy is a fairly unrobust statistic).

(3) To what extent is task-agnosticity a common scenario? Usually the model user is pretty aware of which task they would like to perform, especially if the tasks are different (for example, if you need to do segmentation and detection within a single model). One scenario that might make it difficult to have task ID at inference is if you have a distribution of input data to perform the same task (e.g. digit classification with MNIST + SVHN), but this feels more like a domain adaptation setting to me. Can the authors expand on why task agnosticity is widely practical, and have the authors considered comparing to some domain adaptation baselines?

(4) In the section on OOD detection with colored MNIST, why was the task-agnostic version of LMC not tested? It seems to detract from one of the central points of the paper to only analyze this setting with task-id-required LMC.

(5) Some discussion on how the z’ threshold is set and how robust results are to this hyperparameter would be welcome.

**Post Rebuttal**: Thank you to the authors for the detailed response. My questions/concerns were largely addressed. I think it is an interesting paper with good evidence backing up claims. I will keep my rating a WA (6).


**Time Spent Reviewing:**

3

---

> ### Author Response · Authors · 2021-08-10
> **Responce to reviewer eYQc**
>
> We thank the reviewer for their time and useful comments.
>
> We have answered their question below. In the answers we provide a concise description of our contribution in the current continual-learning literature. We also clarify and re-stress the importance of task-agnosticity in continual learning. Further, we ran experiments with two new baselines (if you have others in mind please let us know). We will include these into the next version of the paper.
>
> We will reply to any follow-up questions or comments you may have.
>
> *“(1) … So is the main contribution of the paper just the combination of the task-agnosticity with the particular formulation of the module-level training? I’d like to understand in a more concise manner the core impact that the work has from a fundamental science perspective.”*
>
> - The main contribution is that we demonstrate that task agnosticity arises from (local) structural learning that is module-specific as opposed to task- and/or model-specific as in previous works [37,64,49]. We show that local structural learning permits OoD generalization (Sec.4.2) as well as the ability to combine independently trained models in a third model without fine-tuning (Sec. 4.3). Importantly, the OoD generalization is only possible if the agent is task-agnostic in the module selection process, since OoD tasks were not observed at training, the learner has to interpolate between the learned tasks and task-id is of no use. All in all, we believe that a promising direction for future work is developing better local structure learning strategies for modular architectures for CL and other paradigms (e.g. meta-learning). One interesting idea in the context of CL could be also implementing a module specific (latent) generative replay strategy.
>
> *"(3) To what extent is task-agnosticity a common scenario? Usually the model user is pretty aware of which task they would like to perform, especially if the tasks are different (for example, if you need to do segmentation and detection within a single model)."*
> - Superficially, it might seem like the reliance on task-ID isn’t of particular importance, especially in an hypothetical situation where one task is segmentation and the other detection. However, task-agnosticity is of the utmost importance if we strive to develop autonomous continually-learning systems [Aljundi et al., 2019, 79, 12] .
>
> - In incremental classification, for example, relying on task-ID means that the model doesn’t need to perform across-task discrimination. In a simple example, an agent learning about dolphins in the context of learning about marine animals couldn’t differentiate a dolphin from a cat if an oracle doesn’t provide it with the information that a dolphin is a marine animal.
>
> - The analogue restricts reinforcement learning agents to a greater extent. A robot learning to sprint in a straight line in task 1 and to walk in a circle would be incapable of sprinting in a circle, as it is neither task 1 or task 2 but an interpolation between previously observed tasks. In the context of accumulating and composing skills, task-agnosticity is fundamental.
>
> - Now, modularity has long been believed to be an important characteristic for CL methods as they hold the promise for achieving positive forward and backward transfer [58, Ballard et al.,1987]. This idea has made a resurgence in CL lately [64,49]. However, the recently proposed methods cannot be used as a basis for  developing autonomous CL agents as they rely on task-ID. One contribution of our work is thus a tool that elevates modular continual-learning methods one level closer to fully autonomous CL systems. We will add this discussion in the next version of this paper.
>
> *"One scenario that might make it difficult to have task ID at inference is if you have a distribution of input data to perform the same task (e.g. digit classification with MNIST + SVHN), but this feels more like a domain adaptation setting to me. Can the authors expand on why task agnosticity is widely practical, and have the authors considered comparing to some domain adaptation baselines?"*
> - Continual domain adaptation is a scenario in which the output space of the model is identical for all tasks (e.g. as is the case in the permuted-MNIST benchmark,  where the output distribution does not change), hence no task-inference needs to be performed. On the other hand, in task- and class-incremental learning the output space of the model has to be expanded as new tasks are seen: e.g. if the first task is MNIST classification and the second task is CIFAR-100 classification there is no reason to expect the 100-way output layer trained on CIFAR to perform well on 10-way MNIST classification learned before. Hence, a separate output head per task is usually used. Task-incremental learning usually assumes task-id to be available at test time to select the correct output head (Farquhar et al.,2018,Van de Ven et al.,2019, Hsu et al.,2018)
> - In this work we focused on task-ID agnostic incremental learning (aka class-incremental) as it is a more challenging scenario, since in domain-incremental no explicit task-inference needs to be performed. We did consider testing LMC on domain adaptation and comparing it to the mixture of expert methods. We shall fill this gap in the future work. For now, the closest experiment to domain adaptation is the filtered miniImageNet experiment presented in the Appendix E.3.
>
> *"(2) Why do the authors not compare to other established task-agnostic baselines?"*
> - We use experience replay as the main task-agnostic baseline. To out knowledge experience replay based solutions are considered to be the method of choice for task-agnostic CL [Masana et al.,2020]
> - Additionally, we add the OvA-Inn [30] baseline (adopted for task-incremental learning) with a fixed imagenet-pretrained encoder ResNet-18 and task-inference procedure similar to the one used by LMC (the used encoder has 3 times more parameters than LMC) and get the following results in terms of average accuracy (A): $S^{in}$=55.93%, $S^{-}$=56.27% ,$S^{+}$=55.83% , $S^{out}$=49.11% , $S^{pl}$= 57.34%. LMC strongly outperforms this baseline, which can be mainly attributed to the fact that LMC’s modular feature extractor is free to adapt to new tasks by adding new modules when needed.
> - We also use OvA-Inn with non-frozen encoder + EWC [36] to prevent drift in the encoder (OvA-Inn + EWC), for this baseline we could not get beyond ~30% accuracy for all the streams.
> We will include these results in the next version of the paper.
>
> *"The comparisons are mostly to heavily modified versions of task-id-required methods, and these modifications are not too convincing (for example, using MNTDP with entropy is a bit dicey as entropy is a fairly unrobust statistic)."*
> - We have run additional experiments where the task-id of MNTDP was selected based on the largest value of the logits as alternative to the entropy based selection, this baseline also could not attain good results (comparable to the entropy-based task inference). We are not aware of other available task-agnostic baselines to compare to in fair conditions, we will kindly include additional task-agnostic baselines that the reviewer suggests.
>
> *"(4) In the section on OOD detection with colored MNIST, why was the task-agnostic version of LMC not tested? It seems to detract from one of the central points of the paper to only analyze this setting with task-id-required LMC."*
> - Because the module selection at test-time is always task agnostic in case of LMC, the goal of the OoD experiment is to demonstrate how the proposed automatic module selection mechanism enables OoD generalization in isolation. In this experiment we used task-ID only to select the output head (e.g. for the OoD task 0vs.1 Green-Black we select the output head corresponding to the seen task 0vs.1 Red-Black). Since MNTDP is not task-id agnostic by design, using the task-id agnostic version of LMC would give possibility for error introduced through incorrect output head selection, leading to unfair comparison.
>
> *"(5) Some discussion on how the z’ threshold is set and how robust results are to this hyperparameter would be welcome."*
> - The hyperparameter selection is discussed in the Appendix (e.g. B2,D): we use stream-level validation to select the best value of the threshold z using validation accuracy as a metric.
> - To see how the z threshold influences the final number of modules (M) and the average validation accuracy over all seen tasks (A) on 3 streams. We first select runs from the corresponding hyperparameter sweeps that resulted in reasonable performance (i.e. (A)>60%) and report (A) and (M) in Table 1 averaged over the selected runs for different values of the threshold z (hereby the random seed is fixed while other hyperparameters are verified). We observe the following tendency: higher z tends to result in a smaller number of modules yet also smaller validation accuracy.
> - Additionally, in Table 2 we report the (A) and (M) for 3 streams with fixed hyperparameters while only varying z. We highlight the runs which achieved comparable performance as MNTDP with less or comparable number of modules.
>
> Sources:
>
> Aljundi, Rahaf, Klaas Kelchtermans, and Tinne Tuytelaars. "Task-free continual learning." Proceedings of the IEEE/CVF Conference on Computer Vision and Pattern Recognition. 2019.
>
> Farquhar, Sebastian, and Yarin Gal. "Towards robust evaluations of continual learning." arXiv preprint arXiv:1805.09733 (2018).
>
> Van de Ven, Gido M., and Andreas S. Tolias. "Three scenarios for continual learning." arXiv preprint arXiv:1904.07734 (2019).
>
> Ballard, Dana H. "Modular learning in neural networks." AAAI. Vol. 647. 1987.
>
> Masana, Marc, et al. "Class-incremental learning: survey and performance evaluation." arXiv preprint arXiv:2010.15277 (2020).
>
> Hsu, Yen-Chang, et al. "Re-evaluating continual learning scenarios: A categorization and case for strong baselines." (2018).

---

> > ### Author Response · Authors · 2021-08-10
> > **Responce to reviewer eYQc (Part II)**
> >
> > Table 1: (A) and (M) averaged over the selected runs in the hyperparameter sweep ((A)>60%) for different values of the threshold z.
> >
> > | $S^{plus}$ |      |     |   | $S^{in}$ |      |   |
> > |----------|------|-----|---|--------|------|---|
> > | (A)      | (M)  | z   |   | (A)    | (M)  | z |
> > | 64.0     | 21.5 | 2   |   | 66.18  | 19.8 | 4 |
> > | 62.3     | 20.3 | 3   |   | 66.12  | 22.7 | 2 |
> > | 61.5     | 22.7 | 4.5 |   | 65.64  | 19.2 | 5 |
> > | 61.2     | 19.2 | 4   |   | 65.52  | 21.5 | 3 |
> > |          |      |     |   | 65.0   | 15.3 | 8 |
> >
> > Table 2: (A) and (M) with fixed hyperparameters while only varying z.
> >
> > |       |   $S^{in}$ |      |     |   | $S^{-}$ |      |     |   | $S^{+}$ |      |     |
> > |-------|---------:|------|-----|---|-----------|------|-----|---|----------|------|-----|
> > |       |      (A) | (M)  | z   |   | (A)       | (M)  | z   |   | (A)      | (M)  | z   |
> > | MNTDP | 66.4±0.9 | 15.0 |     |   | 65.8±0.6  | 14.6 |     |   | 62.6±1.0 | 21.5 |     |
> > |       |          |      |     |   |           |      |     |   |          |      |     |
> > |       |     72.2 |   24 | 1.5 |   |   **68.54**   |   **15** |   **4** |   |    **62.92** |  **22** | **2.5** |
> > |       |    71.93 |   23 |   1 |   |   67.79   |   15 | 4.5 |   |    58.84 |   22 | 1.5 |
> > |       |    70.67 |   21 |   4 |   |   67.73   |   17 | 3.5 |   |    58.72 |   21 |   2 |
> > |       |    70.45 |   22 |   2 |   |   67.14   |   19 |   2 |   |    58.54 |   17 | 4.5 |
> > |       |     70.2 |   21 | 2.5 |   |   66.93   |   17 |   3 |   |    57.72 |   24 |   1 |
> > |       |     70.1 |   19 | 5.5 |   |   65.97   |   22 |   1 |   |    55.98 |   14 |   6 |
> > |       |    69.39 |   20 | 4.5 |   |   65.52   |   22 | 1.5 |   |    58.71 |   14 | 5.5 |
> > |       |    69.23 |   19 | 3.5 |   |   65.52   |   13 |   5 |   |    55.46 |   14 |   7 |
> > |       |    69.18 |   19 |   5 |   |    **64.5**   |   **13** | **5.5** |   |    54.73 |   16 |   4 |
> > |       |    67.73 |   20 |   3 |   |   64.39   |   12 |   8 |   |    54.72 |    9 | 7.5 |
> > |       |    67.23 |   16 |   8 |   |   64.38   |   13 |   6 |   |    54.56 |   19 | 3.5 |
> > |       |    67.01 |   19 |  10 |   |   64.14   |   13 | 6.5 |   |    54.43 |   14 | 6.5 |
> > |       |    66.87 |   16 |   9 |   |   64.14   |   17 | 2.5 |   |    54.22 |   10 |   8 |
> > |       |    **66.82** |   **15** |   **7** |   |   63.87   |   10 |  10 |   |    54.17 |   10 | 8.5 |
> > |       |    66.58 |   16 | 7.5 |   |   63.79   |   11 |   7 |   |    53.96 |    9 |   9 |
> > |       |    66.57 |   16 | 6.5 |   |   63.68   |   11 |   9 |   |    53.48 |   14 |   5 |
> > |       |    66.35 |   17 | 8.5 |   |   62.13   |   11 | 7.5 |   |    52.33 |   19 |   3 |
> > |       |    65.79 |   17 |   6 |   |   61.95   |   11 | 8.5 |   |    44.57 |   11 |   8 |

---

### Official Review · Reviewer_Vtim · 2021-07-09

**Rating:** 6
**Confidence:** 4

**Summary:**

This paper tries to tackle continual learning with a modular approach and proposes the local module composition (LMC) approach. LMC composes modules dynamically by computing local relevance scores for each layer, computed with the input. This architecture makes LMC perform well for OOD tasks as well as outperforming existing approaches on a continual learning benchmark.

**Limitations And Societal Impact:**

Yes, they are included.

**Main Review:**

Originality & Significance
- Although modular approach is not a new concept in continual learning, LMC allows a soft-gating algorithm that is input-dependent rather than task-dependent. Combining modular expansion with the gating algorithm is the primary contribution of the work in my opinion. The gating algorithm is primarily based on the work of Hocquet et al. [30] and Aljundi et al. [2]. The results demonstrate how LMC can be useful for OOD tasks, but in the usual CL case where task identities are known, it does not show significantly superior performance over other methods.  Overall, this combination of modular architecture with the gating algorithm leads to some interesting results.

Quality
- This paper proposes a new modular approach for CL and shows outstanding performance especially for OOD tasks which I find interesting.
- One of the main benefits of modular approaches is how certain modules can be reused for quite different tasks enabling better parameter sharing and generalization. By looking at Appendix B.5, it seems modules typically specialize into different tasks without much reuse except for very related tasks. An analysis on this would be beneficial.
- This leads to a question, can the superior performance of LMC versus other baseline models be attributed to more trainable parameters? - it is hard to judge without knowing what kind of architectures/parameter counts were used for baseline models.
-  The l2-norm structural objective function could cause dead modules in principle - certain modules may always output 0 and not used, for example at some layer, out of N available modules, only one might be always chosen if N-1 modules are optimized to always output 0s. This could limit the capacity of the network even though the whole network has a large amount of trainable parameters. Was this observed in experiments or how was this avoided?
- In my opinion, the major benefit of modular approach in continual learning is how it does not forget when given a lot of different tasks. However, only five tasks are considered in this work. It would be great to consider the S^{long} setting as in MNTDP paper which this work compares with the most.
- LMC has the additional step of adapting to a new dataset through the projection phase - which I like. To demonstrate the usefulness of the whole architecture, it would be good to see if it can significantly outperform MNTDP when MNTDP is given a chance of adaptation too - for example by having a normalization layer for the new dataset to match the statistics of the previously seen datasets.
- Figure 3 has the same tables for (a) and (d) and this should not be the case.

Clarity
- The paper is well-organized and written. It was easy to read

**Time Spent Reviewing:**

5

---

> ### Author Response · Authors · 2021-08-10
> **Responce to reviewer Vtim**
>
> We thank the reviewer for their time and useful comments.
>
> We have answered their question below and include a discussion on the importance of task-agnosticity. We will ensure to clarify all of these points in the final version of this work. Further, we ran experiments on longer streams and obtained satisfactory results which we report and discuss below.
>
> Please let us know if you have any follow-up questions or comments.
>
> *“The results demonstrate how LMC can be useful for OOD tasks, but in the usual CL case where task identities are known, it does not show significantly superior performance over other methods.”*
>
> - We would like to point out that despite its popularity among researchers “the usual” task-incremental setting with known task identities is less general and less challenging than the setting tackled by LMC. Is it of utmost importance to tackle the task-ID agnostic setting if the goal is to design truly autonomous systems (we kindly refer to a related discussion in the second answer to reviewer 3). Furthermore, catastrophic forgetting, which is still at the center of continual-learning research, can be trivially solved when task identities are known via a *duplicate and fine-tune* strategy [45,]. It is also important to point out that OoD generalization is not possible for methods that rely on task-identities in the module selection process, since OoD tasks that have not been seen during training and require the model to interpolate between previously seen tasks. As pointed out by the reviewer 1, by tackling task-ID agnostic setting LMC brings modular continual-learning methods on a level closer to autonomous and general continual-learning systems.
>
> *“By looking at Appendix B.5, it seems modules typically specialize into different tasks without much reuse except for very related tasks. An analysis on this would be beneficial.”*
> - In Appendix B5 we clearly observe reuse of modules. Hence in Fig.6(a) modules learned for T0 are reused for all the other tasks as indicated by light blue color highlighting several modules at a layer; in Fig.6(b) T0’s modules are reused for T5 and T1’s modules are reused for T3; in Fig.6(c) T1 shares modules with T2, T3 and T4; similarly we see modules being shared in Fig.6(d), yet to a less degree.
> - We also note that tasks T2, T3, T4, and T5, in the 4 CTrL streams are unrelated distractor tasks as intended by the authors of the benchmark [73], which indicates that the fact that LMC spawned new modules when learning these tasks is the correct behavior.
>
> *“This leads to a question, can the superior performance of LMC versus other baseline models be attributed to more trainable parameters? - it is hard to judge without knowing what kind of architectures/parameter counts were used for baseline models.”*
>
> - We first clarify the used architecture for the baseline methods: we describe the used architecture and the number of parameters per module (the functional and the structural part) in Appendix A.4. All baselines were initialized with one module per layer, s.t. the number of parameters in each module is equivalent to the number of functional parameters in a single LMC module. For the baselines that support growing, each new module contains the number of parameters of a functional part of the LMC module.
> - To answer the reviewer’s question, we first note that different methods choose how/when they grow and so the number of parameters of a method after training is part of its “behavior”. A higher performance of LMC on the used streams can potentially be attributed to the higher number of functional parameters resulting from the automatic network expansion (indeed LMC tends to have more modules than e.g. MNTDP in Table 1). However, this also showcases the advantage of LMC in that it is able to add new modules (while still partially reusing knowledge from existing ones due to soft-gating) in a way that maximizes the overall accuracy as compared to baselines
> - Finally, in Table 2 in the response to reviewer 3 we show the average test accuracy (A) for different values of the hyperparameter z (module addition threshold). We highlight in black the runs that achieve better or comparable performance as MNTDP with less or comparable number of instantiated modules. In these highlighted runs LMC does not have the advantage of having more trainable functional parameters, yet is still capable of achieving performance of MNTDP.
>
> *“The l2-norm structural objective function could cause dead modules in principle - certain modules may always output 0 and not used, for example at some layer, out of N available modules, only one might be always chosen if N-1 modules are optimized to always output 0s. This could limit the capacity of the network even though the whole network has a large amount of trainable parameters. Was this observed in experiments or how was this avoided?”*
>
> - The described effect was observed in preliminary experiments in an offline setting (for both invertible l2-norm based and the autoencoder structural components). However, in the continual settings this effect did not occur because of implicit task-supervision that is “naturally” provided by the task-boundaries during training. As detailed in L136-137 and Figure 2, the existing modules are always fixed on task switches (they can be reused but not changed by a new task), hence only the newly added modules will be optimized for a new task at each layer. In this way, modules are automatically trained to specialize on tasks. In contrast, in a multi-task learning regime (offline) where no task information is provided, for modules to specialize on separate tasks the algorithm would be required to perform task inference  [37]. As mentioned in L351-352 studying this and other potential benefits of the continual setting for modularity and compositionality constitute an interesting direction for future work.
>
> *“In my opinion, the major benefit of modular approach in continual learning is how it does not forget when given a lot of different tasks. However, only five tasks are considered in this work. It would be great to consider the S^{long} setting as in MNTDP paper which this work compares with the most.”*
>
> - We will include results on $S^{long}$ in the next version of the paper. Running experiments on this longer stream is computationally expensive (due to hyperparameter tuning) and could not be done during the rebuttal week.
> - For now, we ran experiments on a sequence of 30 tasks (a subset of the $S^{long}$ stream). We get the following average test accuracy (A) and total number of modules (M): **MNTDP** (A)=64.58, (M)=64; **LMC**: (A)=62.44, (M)=50. The fact that LMC almost matches the performance of MNTDP, in which module selection is performed by a task-ID aware oracle, shows that the automatic module selection strategy adopted by LMC is still effective in the presence of a large number of modules. We find this result encouraging.
> - Additionally, we observe that LMC on average achieves better performance with solutions that add less modules, since module selection is “easier” with less modules per layer. We demonstrate this by reporting the statistics of the distribution (mean+-std) of (M) and (A) for the hyperparameter sweeps for both LMC and MNTDP on this 30-tasks stream (we only consider runs with (A)>60%): MNTDP (A) = 63.6+-2.5  (M)=64.3+-10.8, LMC (A) = 61.2+-1.2 (M) = 37.6+-14.5  (e.g. LMC reaches (A)=63.4 with 27 modules, yet we do not report it above due to low validation accuracy)
> - We point out that $S^{long}$ (with 100 streams) is a challenging stream for LMC due to the higher number of modules that have to be instantiated --- a situation in which module selection becomes harder for LMC. Yet LMC tackles a more general setting in which modules are selected without task-id.
>
> *“LMC has the additional step of adapting to a new dataset through the projection phase - which I like. To demonstrate the usefulness of the whole architecture, it would be good to see if it can significantly outperform MNTDP when MNTDP is given a chance of adaptation too”*
>
> - We agree that it can be useful to evaluate MNTDP’s performance when endowed with an ability to learn the newly added modules in a way that maximizes the reusability of other modules on other layers (similar to what is aimed through the LMC’s projection phase). However, as we detail below, we do not know how exactly this could be done.
>
> *“... - for example by having a normalization layer for the new dataset to match the statistics of the previously seen datasets.*
> - We thank the reviewer for the suggestion. Importantly, all modules already include batch normalization layers whose statistics are fixed after the module-specific task was learned (see related discussion in appendix A5)
> - To address the reviewer’s suggestion we ran the following additional experiment with MNTDP: we forward-pass 200 batches of unlabeled samples for each new task before learning this task while keeping batchnorms in the “train” mode. This lets the batch norms capture the statistics of the new task. We repeat this “warm-up” procedure before testing on each task using the task’s test data. We could not observe any improvement in terms of accuracy nor in terms of the total number of modules (e.g. on $S^{-}$ we got (A)=65.95%, (M)=15).
>
> *“Figure 3 has the same tables for (a) and (d) and this should not be the case.”*
> - We thank the reviewer for pointing out the mistake in Figure 3. We will replace figure d) with the correct file, this does not change the overall conclusion.

---

> > ### Comment · Reviewer_Vtim · 2021-08-22
> > **Thank you for the response.**
> >
> > I thank the authors for their detailed response and additional experiments.
> > The response resolved most of my concerns and thus I revise my rating accordingly.

---

### Official Review · Reviewer_KtPP · 2021-07-17

**Rating:** 6
**Confidence:** 3

**Summary:**

This paper proposes a continual learning method by learning module networks and a way to compose them. One notable difference is that the proposed approach does not reply on an external task id as the input. Instead, it learns to predict whether the input batch can be composed with the current modules or need to add another module. The authors show several experiments to validate the performance.

**Limitations And Societal Impact:**

I didn't find any potential negative societal impact of this work.

**Main Review:**

This proposed method serves as an important step toward a more general continual learning setting where task identity is unknown. And the idea is pretty simple and reasonable: by tracking the mean and variance of the structural score output by each module, the model itself could predict whether new module is needed.

However, I had a hard time understanding the key part of the proposed method, which makes me less confident about such an assessment. Specifically, I’m confused about section 3.2 and appendix A.2. Could the authors clarify how the projection and accumulation work? When does the projection and accumulation happen?
- Does that happen at every iteration?
- If not, does that only happen during a new task is added (or during an outlier z score detected)? If so, in figure 2, why the projection only happen in the last layer for t=1 but happen two times for t=3 (I think it’s typo and should be t=2)?
- And what is the output space of the whole network? For example, in the CTrL benchmark, is the final layer of your network a 5-way classifier? If so, does that mean the expansion of final layer is not according to the outlier detection but the task identity?

Another weakness is that the method only shows the continual transfer learning in MNIST. In contrast, the recent method (such as [63]) do the same task on CIFAR and miniImageNet. I would highly encourage the author do more experiments on larger scale and more complex datasets.

Overall, I think this is an important methodology improvement. However, I don’t think I fully understand the details so I will keep my score as borderline for now.

**Time Spent Reviewing:**

4.0 hours

---

> ### Author Response · Authors · 2021-08-10
> **Responce to reviewer KtPP**
>
> We thank the reviewer for their time and useful comments (including catching an important typo, see below).  Please note that we do have transfer experiments on CIFAR-10 and miniImageNet in our original submission.  We have answered their question below and will make sure these points are clarified in the final version of this work.  Please let us know if you have any follow-up questions or comments.
>
> *“Could the authors clarify how the projection and accumulation work? When does the projection and accumulation happen? Does that happen at every iteration?”*
> - Projection and accumulation is triggered every time a new module is added to the learner (i.e. every time an outlier in the learned representations is detected). Outliers can be detected at each layer at each iteration (during the projection phase outlier detection is not performed), hence the projection phase can happen at most L times per task, where L is the total number of layers.
>
> *“why the projection only happen in the last layer for t=1 but happen two times for t=3 (I think it’s typo and should be t=2)?”*
>
> - We thank the reviewer for pointing out the typo! It should indeed be task t=2 instead of t=3. In Figure 2 the projection phase only happens in the last layer for t=1 but two times for t=2 because in that example we assumed that in t=1 only one outlier was detected and in t=2 two outliers were detected (at layer 1 and 2). This was an arbitrary choice to exemplify how projection phases are triggered with the addition of new modules.
>
> *“And what is the output space of the whole network? For example, in the CTrL benchmark, is the final layer of your network a 5-way classifier?*
>
> - The output space of the network equals the output space of the inferred task (for the CTrL streams all tasks are 10-way classification tasks, hence the output space is always 10-way). We use a task-specific output head in the final layer in most baselines except for the task-agnostic experience replay, which uses a single head (i.e. the size of the output space is the sum of the output spaces of all tasks seen).
>
> *"If so, does that mean the expansion of final layer is not according to the outlier detection but the task identity?”*
>
> - During training, output modules at the final layer are added every time a new task arrives --- i.e. at task boundaries (this is mentioned in L105-106). LMC can potentially be extended to a setting where task boundaries are not observed during training through e.g. applying the proposed outlier-detection mechanism also to the output layer.
>
> *“Another weakness is that the method only shows the continual transfer learning in MNIST. In contrast, the recent method (such as [63]) do the same task on CIFAR and miniImageNet.”*
>
> - We provide experiments on CIFAR and miniImageNet datasets. As described in Section 4.1 and detailed in Appendix B, the CTrL benchmark includes the CIFAR-10 dataset. Moreover, it is the dataset for which the transfer is actually measured in the $S^{+}$, $S^{-}$ and $S^{out}$ streams, since CIFAR is the first and the last task in those streams. MiniImageNet results for the continual meta-learning setting are provided in Appendix E.3 as mentioned in l.302.

---

> > ### Comment · Reviewer_KtPP · 2021-09-09
> > **Thanks for your response**
> >
> > Thank you for your detailed explanation. I think most of my concerns are resolved.

---

### Official Review · Reviewer_Hxbt · 2021-09-12

**Rating:** 7
**Confidence:** 5

**Summary:**

The paper proposes a new algorithm for addressing continual learning (CL) by leveraging modular architectures. In contrast to (the few) existing modular CL works, this one focuses on the task-agnostic setting, where the agent is not informed of the task ID. The proposed architecture assigns two components to each module: a functional component and a structural component. The latter is in charge of locally determining whether each data point (actually each mini-batch, in practice) is in-distribution for the given module. The training proceeds in stages, adding new modules whenever the new data point is out-of-distribution for all modules, then training the module such that its output is in-distribution for modules at later depths, and finally training the entire model together. Extensive experiments are carried out to demonstrate the strengths of the proposed approach.

**Limitations And Societal Impact:**

Yes.

**Main Review:**

############## Strengths ##############

1. The problem setting is highly relevant and vastly understudied
2. The proposed approach addresses one particular shortcoming of existing modular CL methods: the ability to work without task IDs
3. The experiments are extensive, comprehensive, and well described and analyzed

############## Weaknesses ##############

1. The description of the technical approach is too short and somewhat hard to follow, making several choices unclear and unjustified
2. The continual meta-learning experiments seem disconnected from the rest of the paper to me
3. The method has no mechanism for limiting the amount of experts that are added, which is empirically demonstrated to lead to a large number of modules in the only experiment where this is evaluated

############## Recommendation ##############

This is a solid submission and I recommend accepting it. While technically the submission doesn't introduce a fancy new method with many technical contributions, it does demonstrate a clear understanding of the challenges of modular CL, and addresses one of the shortcomings of existing modular CL approaches. To me the biggest plus of the manuscript is the quality of the empirical evaluation, with well-chosen baselines evaluated on a variety of settings.

############## Arguments ##############

On the technical side, the key contribution is proposing for each module to individually select whether each data point should be routed to it. This is a relatively minor technical contribution, but I do not think that is necessarily problematic given that this is a fairly new problem setting. What does concern me is that this section is very short and I don't believe that it is described in enough detail. In particular, no intuition behind the structural loss is given. I make out between the main text and Appendix A.3 that the idea is for this module to serve as a density estimator that detects out-of-distribution samples. However, it is unclear how the losses described in Sec 3 achieve this, or why different losses must be used for the feature extractor and the outputs.

One interesting technical insight is that training the structural component locally and with a separate loss than the prediction loss enables fully-differentiable training even if using hard modular selections. The authors also clearly explain that the key challenge for learning a task-agnostic modular component is that this component itself can suffer from forgetting if using standard gating functions, which is not the case with the local approach proposed here. The projection phase is an interesting idea to ensure that the new module is compatible with other modules, and this is possible also due to the choice of local module selection.

The empirical evaluation is very strong. The first experiment on CTrL (a benchmark specifically designed to test properties that modular CL methods should exhibit) compares against a wide array of well-chosen baselines, including the two primary related works (Mendez & Eaton and Veniat et al.). Results in Table 1 are strong and convincing. Then the second experiment proposes a benchmark specifically for out-of-distribution generalization, which was not studied in either of the preceding modular CL works, and the authors again demonstrate a strong ability to generalize OOD. The third experiment demonstrates that independently trained models can later be combined to achieve equal-or-better performance than the original separate models.

That being said, I recommend removing the final experiment on continual meta-learning. The problem setting for this experiment is unclear (even though it's somewhat described in Appendix E). But the main problem is that this is a completely different problem setting than studied in the rest of the paper, plus even the algorithm is modified to deal with the modified problem setting. It seems disconnected and contributes to the extreme length of the paper (26 pages including appendices).

One concern which the authors only mention in the conclusion is that the model grows quite rapidly. This is demonstrated in the results of Table 1, where the proposed method uses nearly one new module per task, just like the MoE. While in principle using modular architectures enable sub-linear growth (as the authors mention) it isn't clear whether their proposed method is actually sub-linear in practice. Note that having fewer than one new module per task is _not_ equivalent to being sub-linear (e.g., maybe the model learns 0.5 new modules per task, which is still linear). Part of what makes this such a major issue is that the model is allowed to grow at every sample (or mini-batch), which is quite frequent.

############## Additional feedback ##############

The following points are provided as feedback to hopefully help better shape the submitted manuscript, but did not impact my recommendation in a major way.


Intro
- "Modules self-determine their relevance given an input" -- sounds very similar to the competitive mechanism of RIMs [1]. Perhaps a comparison is warranted.

Sec 3
- Equation 7 doesn't make it clear at all that the structural components are only trained via the structural loss. With soft modularity (as used in most experiments), the first term (the overall loss) is also differentiable w.r.t. the structural parameters, but later it is made clear that this is not used. It might be best to present this as two separate objectives used to train disjoint sets of parameters.

Conclusion
- I like how the conclusion points to actually relevant directions for future work.

Typos
- CtRL is used multiple times, but it's CTrL

Appendices
- The entire meta-CL addition is very long, and I would frankly recommend removing it from the paper. It's a whole new problem setting (which I don't think is described in full detail) with some changes in the algorithm to address this particular problem setting.

[1] Goyal et al. "Recurrent Independent Mechanisms," (2020).

**Time Spent Reviewing:**

3

---

### Author Response · Authors · 2021-09-09
**Minor correction Table 1**

We would like to kindly inform the reviewers about a minor mistake that we have found in Table 1: the mean accuracy $\mathcal{A}$ for LMC(A) in stream $S^-$ (row 12 column 2) was incorrectly copied from the experiment reports and should be $67.2\pm1.5$ instead of the state the $69.0\pm1.5$. This does not change the overall conclusions and we will correct this in the next version of the paper.

---

### Decision · Program_Chairs · 2021-09-27

**Decision:**

Accept (Poster)

**Comment:**

This paper originally received three borderline positive reviews but without high confidence, and so a fourth review was solicited after the regular review period from a reviewer who is extremely knowledgeable on the topic.  From this fourth review, it is clear that this paper should be accepted.

This paper focuses on continual learning of modular representations in task-agnostic settings, develops an approach that addresses this challenging, important, and understudied problem, and performs an extensive empirical analysis on a variety of well-chosen baselines.

The authors should be aware that there are some unclear parts, identified in the reviews, that even the most expert of the reviewers found difficult to follow at times.  These need to be clarified for the final version.